# Modulation of sensory prediction error in Purkinje cells during visual feedback manipulations

Martha L. Streng[1,2], Laurentiu S. Popa[2] & Timothy J. Ebner[1,2]

It is hypothesized that the cerebellum implements a forward internal model that transforms motor commands into predictions about upcoming movements. The predictions are compared with sensory feedback to generate sensory prediction errors critical to controlling movements. The simple spike firing of cerebellar Purkinje cells both lead and lag movement consistent with representations of motor predictions and sensory feedback. This study tests whether this leading and lagging modulation provides the prediction and sensory feedback necessary to compute sensory prediction errors. Two manipulations of the visual feedback are used in rhesus monkeys performing pseudo-random tracking. Consistent with a forward model, delaying the visual feedback demonstrates that the leading simple spike modulation with position error is time-locked to the hand movement. Reducing the feedback shows that the lagged modulation is directly driven by visual inputs. Therefore, Purkinje cell discharge carries both the motor predictions and sensory feedback required of a forward internal model.

[1] Graduate Program in Neuroscience, University of Minnesota, Minneapolis, MN 55455, USA. [2] Department of Neuroscience, University of Minnesota, Minneapolis, MN 55455, USA. Correspondence and requests for materials should be addressed to T.J.E. (email: ebner001@umn.edu)

A fundamental question is how the central nervous system controls complex movements online. A compelling hypothesis is that to control movements, the central nervous system implements a forward internal model that predicts the consequences of motor commands[1–4]. These predictions are then compared to the actual consequences, resulting in a sensory prediction error (SPE) that is used for online control and motor learning[2,3]. Evidence suggests that the cerebellum acquires and implements forward internal models and plays a role in computing SPEs[2,5–13]. However, the mechanisms by which the predictive and feedback components of the forward internal model are represented in the activity of cerebellar neurons have yet to be fully elucidated.

The high-frequency simple spike (SS) discharge of Purkinje cells encodes robust information about movements including performance errors and kinematics and has many properties consistent with the output of a forward internal model (for reviews see refs. [14,15]). Importantly, SS modulation both leads and lags behavior showing that individual Purkinje cells carry predictive and feedback signals about movements[16–19]. The

predictive and feedback modulations have opposing actions consistent with the components of a SPE and these signals can be used to decode both future and past performance errors and kinematics with considerable accuracy[16,20].

It is essential to establish that the feedforward and feedback SS signals of Purkinje cells respond to experimental manipulations as required of a forward internal model. To address this issue, this study assessed how disrupting visual feedback during pseudo-random tracking alters the predictive and feedback SS activity. Visual feedback was either delayed by introducing a lag between manipulandum and cursor movements to test that the predictive SS modulation is time-locked to motor commands or reduced by selectively hiding the cursor while it was inside the target to determine whether the feedback SS modulation is driven by visual error information. Altering the visual feedback shows that for position error the predictive SS modulation is coupled to hand movement while the feedback modulation is driven by sensory feedback. Therefore, the SS activity of individual Purkinje cells independently encodes predictions based on motor commands and sensory feedback about the corresponding behavior.

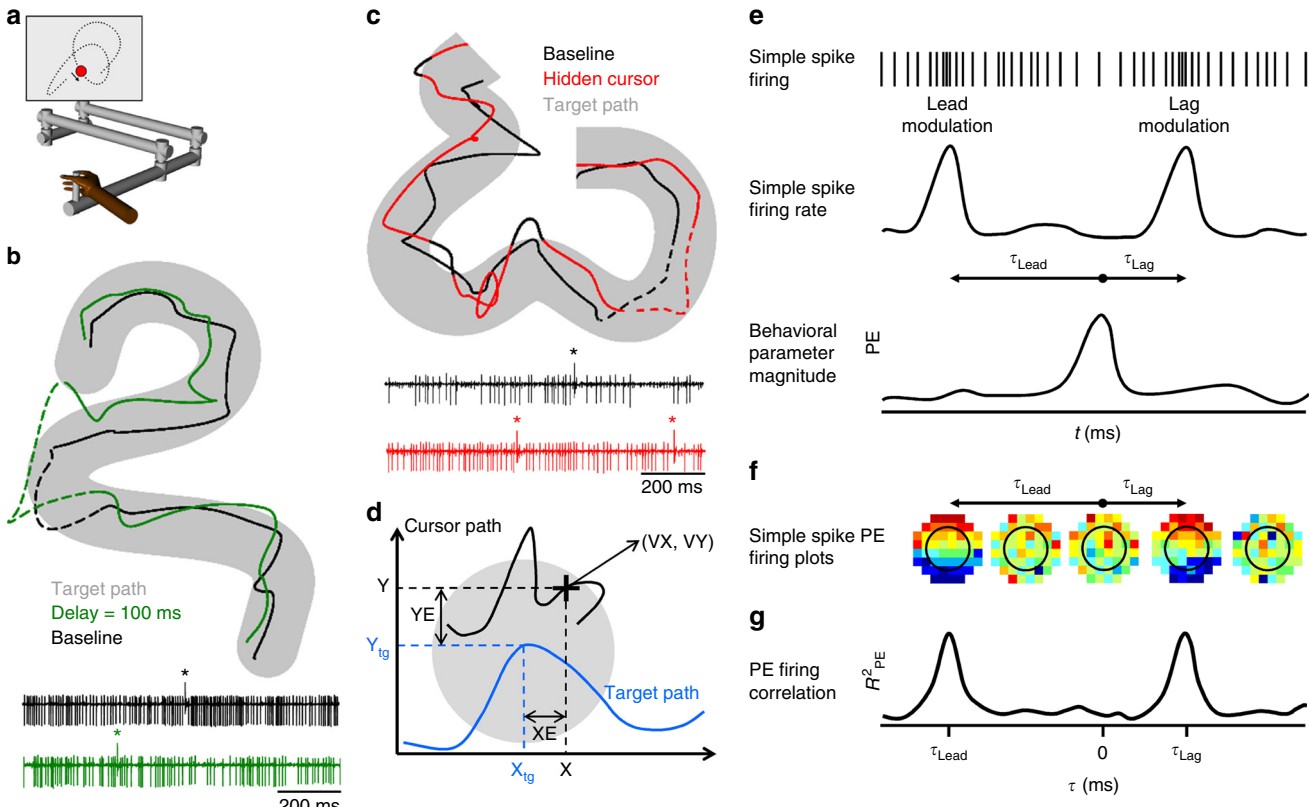

**Fig. 1** Experimental setup, example Purkinje cell recordings during visual feedback manipulations, and analysis steps. **a** Monkeys use a manipulandum to track a circular target (red circle) moving pseudo-randomly (target trajectory depicted by dotted line) on a computer screen. Image modified from Streng et al.[21]. **b** Cursor path in baseline (black trace) and delay conditions (green trace) while the monkey tracked the target moving on the same trajectory (gray area). Example Purkinje cell activity recorded during delay (green) and baseline (black) conditions during the 1 s dashed segments of the trajectories. Complex spikes marked by '*'. **c** Cursor path in baseline (black trace) and hidden (red trace) conditions while the monkey tracked the target moving on the same trajectory (gray area). In the hidden condition, the animal does not see the cursor when it is inside the target (red portion of the cursor movement). The cursor is visible only when it is outside the target (black portion of the cursor movement). Example Purkinje cell activity recorded during baseline (black) and hidden (red) conditions during the 1 s dashed segments of the trajectories. **d** Behavioral parameters analyzed. Target path (gray circle) is depicted by the blue trace. Hand movement is mapped onto the cursor movement (black trace). Movement kinematics are described by position (X, Y) and the velocity vector (VX, VY). Position error (XE, YE) is the difference between cursor position (X, Y) and target center ($X_{tg}$, $Y_{tg}$). **e** Illustration of the analysis steps. Cartoon of simple spike train (top panel) is transformed to a mean-subtracted continuous firing rate (middle panel) and correlated with the behavior parameters (bottom panel). Arrows in the middle panel illustrate the lead and lag relations ($\tau$) between SS modulation and behavior. PE: position error. **f** Example of mean-subtracted SS firing rate across all trials, binned and averaged in 8 × 8 partitions in relation to behavior (PE in this illustration) at both lead and lag $\tau$-values. Red colors indicate higher discharge rates. **g** Temporal linear regression analysis results in $R^2$ as a function of $\tau$ that quantifies the relation between SS activity and behavior

## Results

**Pseudo-random tracking and experimental paradigm.** Monkeys were trained to use a 2-joint manipulandum to track a circular target moving on a computer screen (Fig. 1a). The paradigm employed a set of 100 trajectories in which the target path, speed and duration varied pseudo-randomly within certain constraints (see Materials and Methods). Two variants of the paradigm were studied in which the visual feedback was altered to understand the sources of the lead and lag SS modulation of the Purkinje cells (Fig. 1e) in the context of the forward model hypothesis. The first variant, the delay condition, introduced a 100 or 200-ms delay between the movement of the manipulandum and movement of the cursor (Fig. 1b). The second variant, the hidden cursor condition, reduced visual feedback by only displaying the cursor when it was outside and not when inside the target (Fig. 1c). Purkinje cells, identified by the presence of SSs and complex spikes (Fig. 1b and c), were recorded in the baseline and in one of the altered feedback conditions. In all conditions, the animals were required to keep the cursor within the target and were allowed only brief excursions outside the target (<700 ms) before the trial was aborted.

Analyses evaluated the SS firing with both position error and kinematics. Position error, defined as the difference between the cursor and target positions (XE and YE, Fig. 1d), provides a continuous measure of performance error. Position error is defined relative to the cursor and target movements, not with respect to the hand. Movement kinematics were described by hand position (X and Y) and velocity (VX and VY) (Fig. 1d). Our previous studies established that SS firing modulates with position error as well as kinematics and that SS modulation of individual Purkinje cells can both lead (i.e., predict) and lag (i.e., provide feedback) these parameters (Fig. 1e)[16,21]. To visualize single Purkinje cell SS modulation with behavioral parameters, mean-subtracted SS firing rate across all trials was binned and averaged in $8 \times 8$ partitions of the behavior at both lead and lag $\tau$-values as shown for position error in Fig. 1f. Temporal linear regressions in 20 msec time steps ($\tau$-value) were used to determine the predictive and/or feedback SS modulation with each pair of parameters; position error (XE and YE), position (X and Y) and velocity (VX and VY) (Fig. 1g and see Materials and Methods)[16,22–25]. The temporal linear regressions result in $R^2$ (Fig. 1g) and sensitivity temporal profiles that provide measures of the modulation strength and sensitivity with the parameters, respectively.

**Rationale for the delayed visual feedback paradigm.** The hypothesis tested by these experiments states that the leading SS modulation is due to a forward model that predicts upcoming position errors while the lagging modulation is driven by sensory feedback. Given that the motor commands control hand movements, the output of the model has to make predictions about position error time-locked to the hand movement. In the baseline condition, SS firing encodes predictions of the upcoming position error consistent with a forward model (Fig. 2a). However, because the movements of the hand and cursor are coupled, whether the modulation with position error is relative to the hand or cursor movements cannot be determined (Fig. 2a). Delaying the cursor relative to the hand differentiates the hand and cursor movements (Fig. 2c) and allows testing whether the lead modulation with position error is relative to the hand or cursor. If the lead modulation is based on motor commands, it will be coupled to the hand and will occur earlier relative to the delayed position error (Fig. 2c). The hypothesis also states that the lagging SS modulation with position error is feedback driven and coupled to the cursor movements, therefore, the timing of the SS lag modulation will not change. The effects of delaying the visual feedback will be manifest in the $R^2$ profiles as a shift in the lead peak to more negative $\tau$-values and no change in the timing of the lag peak (Fig. 2b vs. 2d).

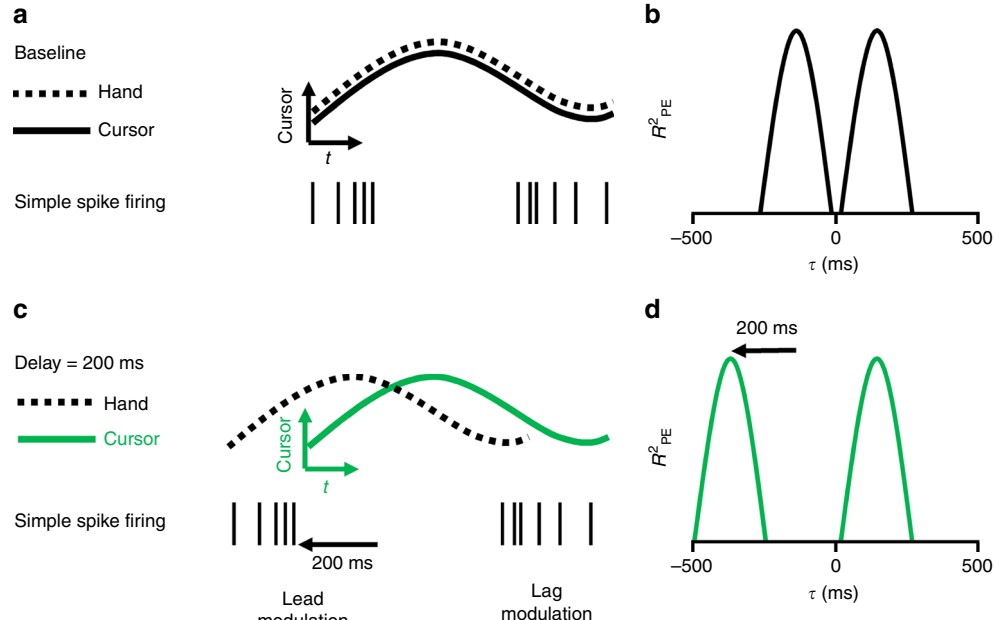

**Fig. 2** Rationale for the delay condition experiment. **a** In the baseline condition the cursor (continuous trace) and hand (dashed trace) movements are indistinguishable. Simple spike firing both leads (left spike train) and lags (right spike train) the cursor movement. **b** Temporal $R^2$ profile of the SS firing with the position error includes a leading ($\tau < 0$) and lagging ($\tau > 0$) peak. **c** In the delay condition, hand movement occurs before the cursor movement by the delay imposed. Based on the forward model hypothesis, the lead SS modulation is time-locked to hand movement and will shift earlier relative to cursor movement. The lag modulation should be time-locked to the cursor movement. **d** In the delay condition, the lead $R^2$ peak should shift earlier and the lag $R^2$ peak should be unchanged

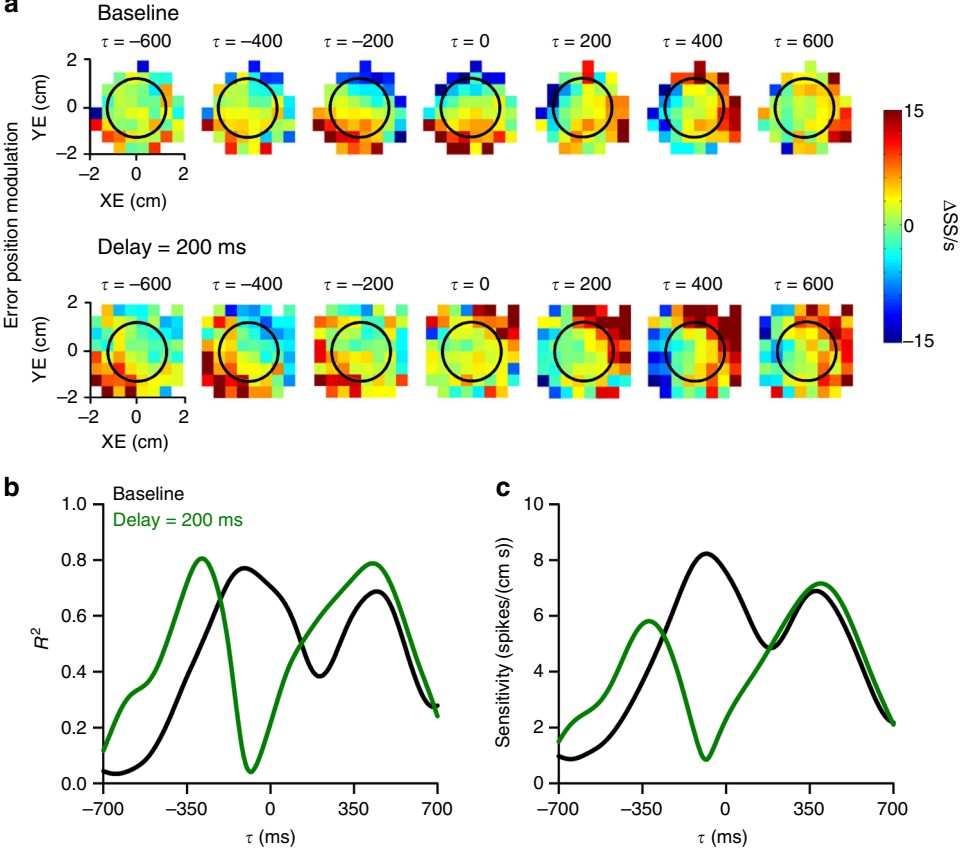

**Fig. 3** Visual feedback delay shifts predictive encoding of position error. **a** Firing maps for an example Purkinje cell with lead and lag SS position error modulation in both baseline (top row) and 200 ms delay (bottom row) conditions. Each map indicates SS modulation at a specific lead (negative $\tau$) or lag (positive $\tau$). Black circle indicates target edge. $R^2$ (**b**) and sensitivity (**c**) temporal profiles for the Purkinje cell shown in **a** computed using linear regression analyses (see Materials and Methods) that quantifies the SS encoding of position error in both the baseline (black line) and 200 ms delay (green line) conditions. As for the firing maps, negative $\tau$ indicates lead SS encoding

**Visual feedback delay shifts lead encoding of position error.** Sixty-two Purkinje cells were recorded from two rhesus macaques during pseudo-random tracking under both baseline and visual feedback delay conditions. Forty three cells were evaluated with a delay of 100 ms and 19 with a delay of 200 ms. Purkinje cells were recorded in lobules IV–VI in the intermediate cerebellar zone in which SS firing is known to modulate with position error and limb kinematics (for reviews see refs.[15,26]). Of this population, 50 Purkinje cells were analyzed further that had significant SS modulation with at least one parameter (eight with position, 38 with velocity, and 30 with position error) in both baseline and delay conditions. Of the 30 neurons modulated with position error, nine had only lead encoding, 12 had only lag encoding, and nine had both lead and lag encoding (see Supplementary Figure 1a and b for the distributions of the encoding properties).

The SS firing plots for an example Purkinje cell recorded during baseline and delay conditions exhibit the characteristic lead and lag modulation with position error observed previously during pseudo-random tracking (Fig. 3a)[16,21]. During baseline, strong SS modulation leads position error with increased firing in the lower half of the workspace. At feedback times, SS modulation is strongest on the right side of the error workspace. When visual feedback is delayed by 200 ms, the lead SS modulation shifts to more negative $\tau$-values in agreement with the hypothesis (see Fig. 2). Conversely, the timing of the lag modulation is unaffected. Linear regression analysis was used to determine the timing and amplitude of the lead and lag SS discharge with position error. For this same Purkinje cell, the $R^2$ and sensitivity temporal

profiles (Fig. 3b and c) confirm that the SS lead and lag peaks at −100 and 440 ms, respectively, during baseline tracking. In the delay condition, the timing of predictive position error encoding occurs earlier, peaking at −280 ms, while the peak timing of the feedback modulation (420 ms) is unaffected, for both the $R^2$ (Fig. 3b) and sensitivity profiles (Fig. 3c).

For the 30 cells with position error encoding, significant shifts in the timing of the lead SS encoding occur for both 100 and 200 ms delays (Fig. 4a and b, 100 ms: $t(10) = 3.18$, $p = 0.01$; 200 ms: $t(6) = 5.15$, $p = 0.002$, paired $t$-tests). The average shift in the lead position error encoding is approximately equal to the duration of the experimental feedback delay ($-118 \pm 123$ ms and $-206 \pm 105$ ms for the 100 and 200 ms delays, respectively) (Fig. 4c). As the hypothesis predicts (Fig. 2), the timing of the lag SS modulation with position error is not affected by the delay (Fig. 4a–c, 100 ms: $t(10) = 0.68$, $p = 0.51$; 200 ms: $t(6) = 1.03$, $p = 0.34$, paired $t$-tests). Importantly, the average encoding strength of position error, both the lead and lag modulations, is not affected by the feedback delay (Fig. 4d, $F(1,79) = 1.31$, $p = 0.27$, ANOVA). Together, these results show that the shift in the leading SS modulation encodes a prediction of position error with respect to the hand and not the cursor, as expected for the output of a forward internal model. The lagging SS modulation is invariant with the delay showing that it is driven mostly by visual feedback with no major proprioceptive contribution.

**Rationale for the visual feedback reduction paradigm.** The second manipulation, the hidden cursor condition, reduced the

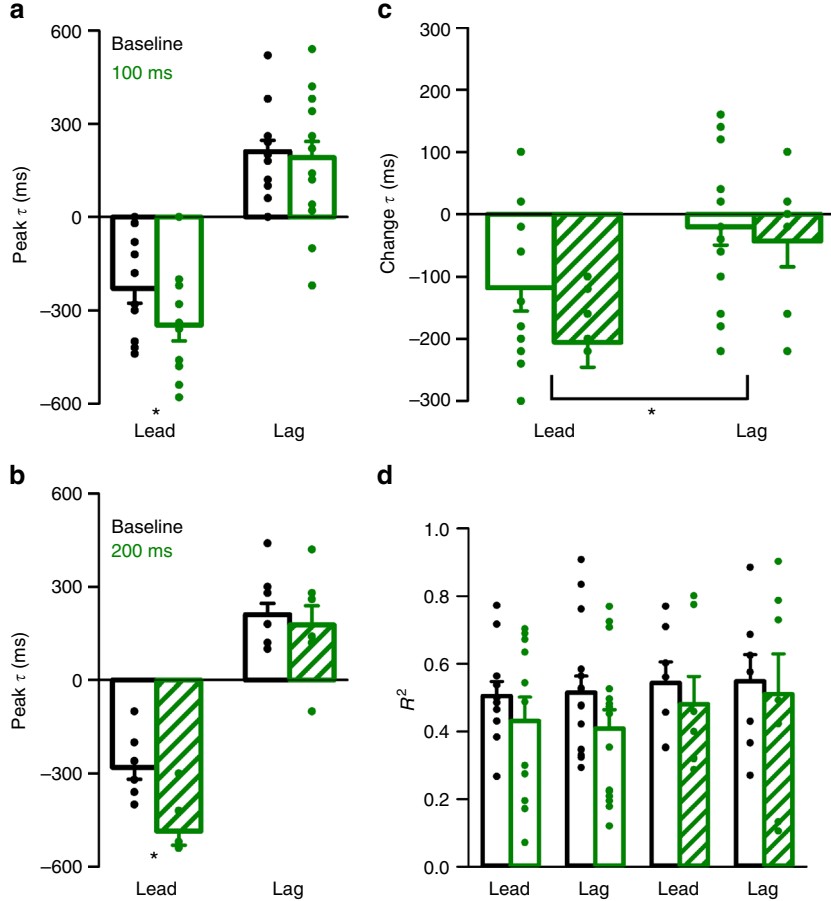

**Fig. 4** Delay condition population results for the position error encoding. **a**, **b** Average peak timing of encoding across the population illustrates the significant shift in timing of lead encoding for both 100 ms (**a**, $t(10) = 3.18$, $p = 0.01$, paired $t$-test) and 200 ms (**b**, $t(6) = 5.15$, $p = 0.002$, paired $t$-test) delays. **c** The time of lag encoding was not significantly affected for either delay ($t(10) = 0.68$, $p = 0.51$ for 100 ms, paired $t$-test and $t(6) = 1.03$, $p = 0.34$ for 200 ms, paired $t$-test). Average change in the timing of lead and lag encoding in the 100 ms (solid green) and 200 ms (checkered green) conditions, illustrating a significant shift in the timing of lead SS encoding ($F(1,39) = 11.732$, $p = 0.002$, ANOVA). **d** Average $R^2$ of lead and lag SS encoding of position error, indicating no significant change in encoding strength for either delay condition ($F(1,79) = 1.31$, $p = 0.27$, ANOVA). Error bars in this and all figures indicate SEM

visual feedback by not displaying the cursor inside the target (Fig. 5). Reducing the visual feedback tests if the lagged SS modulation with position error is driven by the visual feedback. Compared to the baseline condition (Fig. 5a), the modulation should decrease whenever the cursor is inside the target (Fig. 5c). Conversely, if the lead SS encoding is predicting the position errors based on the motor commands, it should be unaffected. In the hidden condition, the $R^2$ temporal profile for the lagging SS modulation with position error is expected to decrease and lead modulation should not change (Fig. 5b vs. 5d).

**Feedback reduction decreases lagged position error encoding**. Thirty-six Purkinje cells were recorded during pseudo-random tracking under both baseline and the hidden cursor conditions. Thirty-three of these cells that had significant encoding of at least one parameter, including 16 correlated with position, 30 with velocity, and 26 with position error (see Supplementary Fig. 1c) were analyzed further. Of the 26 modulated with position error, 11 had lead encoding only, five had lag encoding only, and 10 had both lead and lag encoding.

The hidden cursor condition decreases the feedback SS modulation with position error. This is illustrated for an example Purkinje cell with strong SS modulation in the upper half of the error workspace that leads the behavior, and strong modulation

in the lower half of the error workspace that lags the behavior (Fig. 6a). During the hidden cursor condition, the predictive SS modulation is maintained with increased firing in the upper half of the workspace, both inside and outside the target (black circle). The lagged SS modulation with position error inside the target is markedly reduced. The SS modulation was quantified using linear regressions as for the visual feedback delay. However, the analysis was restricted to the behavior and SS firing inside the target as we reasoned that the lagged SS encoding would be reduced within but not outside the target. The $R^2$ and sensitivity profiles (Fig. 6b and c) mirror the effects observed in the firing maps; the hidden cursor condition greatly reduces the magnitude of feedback SS modulation with position error, but not the predictive SS encoding.

Across the population, feedback SS modulation with position error decreased in the hidden cursor condition (Fig. 7). A significant reduction in the SS modulation in the hidden condition at the single cell level was defined as significant encoding in the baseline but not in the hidden cursor condition (see Materials and Methods). Using this conservative criterion, during the hidden cursor condition the lag encoding of position error decreased in 73% (11/15) of Purkinje cells. Accordingly, the average $R^2$ of the lagged SS modulation in these cells decreased significantly with no change in the average $R^2$ of the lead

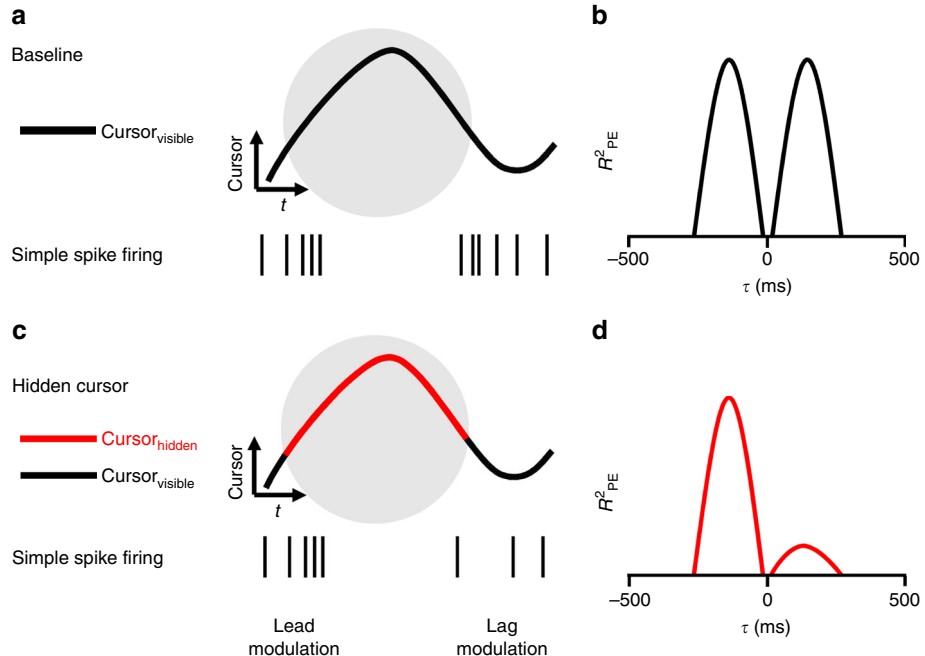

**Fig. 5** Rationale for the hidden cursor experiment and expected results. **a** In the baseline condition, the cursor (black trace) is visible whether inside or outside of the target (gray circle). Simple spike firing both leads (left spike train) and lags (right spike train) the cursor movement. **b** Temporal $R^2$ profile of the SS firing with the position error (PE) includes a leading ($\tau < 0$) and lagging ($\tau > 0$) peak. **c** In the hidden condition, the cursor is visible only when outside the target (black trace) and invisible within the target (red segment). Based on a forward model, the lead SS firing will not be affected, but the lag firing will be reduced. **d** In the hidden condition, the lead $R^2$ peak will be unchanged and the amplitude of the lag $R^2$ peak will be reduced

modulation (Fig. 7a, $F(1,35) = 10.65$, $p = 0.0026$, ANOVA). The four Purkinje cells that maintain lagged encoding also showed an overall decrease (average $R^2$ of $0.79 \pm 0.06$ in baseline and $0.62 \pm 0.12$ in the hidden cursor conditions). Therefore, reduced visual feedback selectively decreases the feedback position error SS encoding in all 15 Purkinje cells. Furthermore, the peak times of the lead and lag modulation are not affected by the hidden cursor (Fig. 7b, ($F(1,45) = 1.04$, $p = 0.31$, ANOVA). These results strongly suggest that the lagged SS modulation represents visual feedback encoding of position errors. Conversely, the lack of change in the lead SS modulation is consistent with an intact predictive signal based on motor commands.

**Feedback manipulations do not affect SS kinematic encoding.** Although the SS firing during pseudo-random tracking is highly correlated with position error, SS firing also modulates with kinematics, including hand position and velocity[16,22]. While position error is defined as the difference between cursor and target positions on the screen, the kinematic measures could potentially reflect the visual kinematics of the cursor as well as the kinematics of limb movement. Therefore, an additional analysis addressed whether the SS modulation with kinematics primarily reflects the encoding of movements of the limb and/or vision of the cursor movements. If the SS modulation with kinematics primarily reflects limb movement information, then the timing and the strength of the lead and lag encoding with hand kinematics should be unaffected in the delay and hidden cursor conditions.

Two example Purkinje cells recorded during the delay and hidden cursor conditions, respectively, are shown in Fig. 8. The first example has strong lead SS encoding of velocity in both baseline and delayed cursor conditions (Fig. 8a–c). Importantly, the timing of both peak encoding strength (Fig. 8b) and sensitivity (Fig. 8c) are unaffected by a delay of 100 ms (note that the profiles are SS modulation with hand velocity, not cursor

velocity). Similar results are observed for the 38 Purkinje cells with significant encoding of position or velocity, with no significant change in the timing of lead or lag encoding of kinematics (Fig. 8d, $F(1,107) = 0.15$, $p = 0.70$, ANOVA) or the strength of the encoding (Fig. 8e, $F(1,107) = 0.47$, $p = 0.71$, ANOVA). The second example cell has strong lead and lag encoding of velocity in baseline and hidden cursor conditions (Fig. 8f–h). As done for the analysis of the position error encoding, the SS modulation with kinematics in both baseline and hidden cursor reflects the encoding only inside the target. Unlike the SS encoding of position error inside the target, the magnitude of the lagged encoding of velocity inside the target is unaffected by the feedback reduction paradigm (Fig. 8g, h). For the 30 Purkinje cells with significant encoding of position or velocity evaluated in the hidden cursor condition, there is no significant change in either lead or lag encoding of kinematics (Fig. 8i, $F(1,113) = 0.01$, $p = 0.93$, ANOVA) or the $R^2$ (Fig. 8j, $F(1,113) = 1.48$, $p = 0.23$, ANOVA). Together, these results show that SS modulation with kinematics during pseudo-random tracking reflects the encoding of limb movements, not the feedback associated with movement of the cursor. Further, the observations demonstrate that the SS position and velocity representations are independent of position error encoding.

**Manipulations of visual feedback and behavior.** During baseline and delay conditions, animals strive to keep the cursor near the target center and there are no significant differences in the probability densities for position (100 ms: $F(1,127) = 0.67$, $p = 0.42$; 200 ms: $F(1,127) = 0.52$, $p = 0.52$), velocity (100 ms: ($F(1,127) = 0.08$, $p = 0.77$; 200 ms: $F(1,127) = 0.12$, $p = 0.73$), and position error (100 ms: $F(1,127) = 0.95$, $p = 0.33$; 200 ms: $F(1,127) = 0.22$, $p = 0.64$) due to the delay (Supplementary Fig. 2a–c). Therefore, for the two conditions the evaluation of the SS firing was assessed over similar movements. In addition, the similar coverage of the workspaces suggests the visual feedback

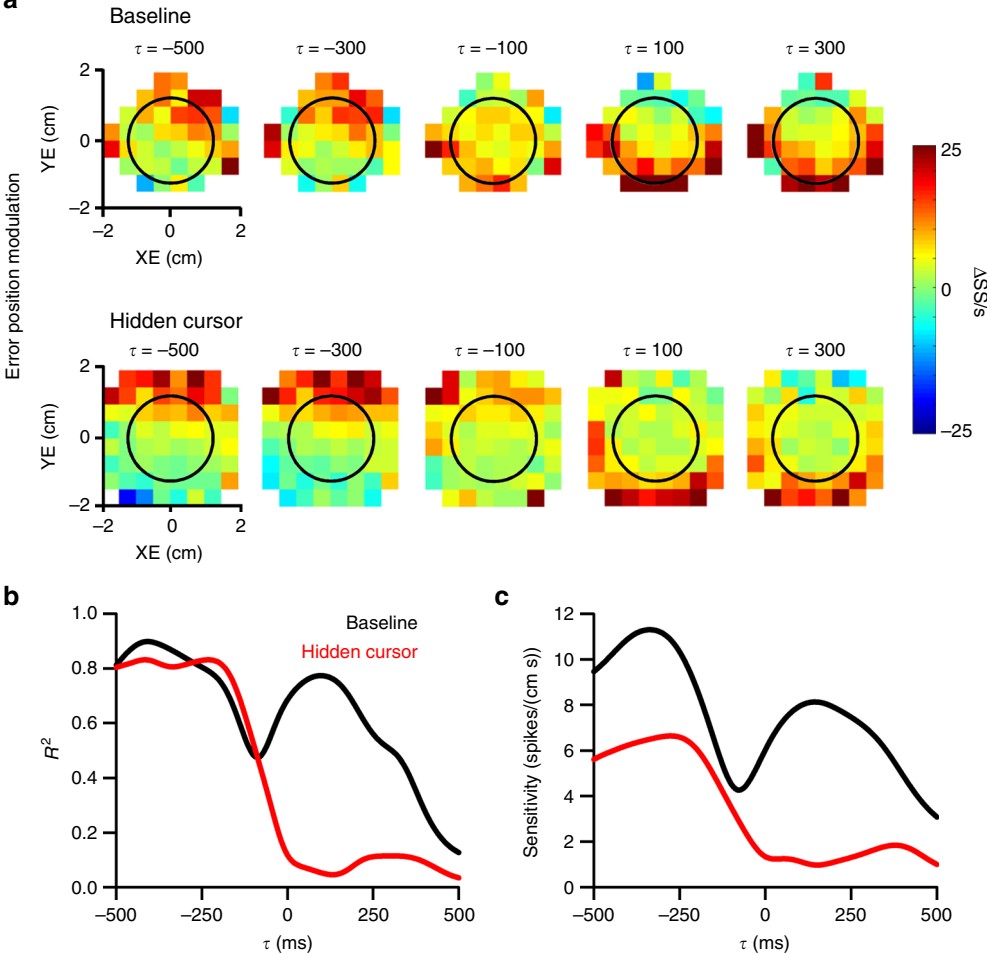

**Fig. 6** Hidden cursor reduces feedback encoding of position error. **a** Firing maps for an example Purkinje cell with lead and lag SS encoding of position error in the baseline (top row) and hidden cursor (bottom row) conditions. Black circle indicates target edge. $R^2$ (**b**) and sensitivity (**c**) profiles for the Purkinje cell in **a** illustrate the decrease in lagged SS encoding of position error

delay does not induce a significant change in tracking strategy. The paradigm was designed to minimize adaptation by removing the visual feedback manipulations between trials and the animals show no evidence for adaptation either across trials (100 ms: $\rho = 0.07$, $p = 0.63$; 200 ms: $\rho = 0.37$, $p = 0.12$) or recording days (100 ms: $\rho = -0.03$, $p = 0.89$; 200 ms: $\rho = 0.47$, $p = 0.0006$) (Supplementary Fig. 2d, e). This likely reflects removal of the feedback manipulations between trials, as rapid switching between task conditions limits adaptation[27]. However, the delay does increase task difficulty as position error magnitude increases (Supplementary Fig. 2f, 100 ms: $t(43) = -4.89$, $p < 0.001$; 200 ms: $t(18) = -12.74$, $p < 0.001$).

Workspace coverage is also similar in both baseline and hidden cursor conditions, with no significant change in the probability densities for position ($F(1,127) = 1.74$, $p = 0.19$, ANOVA), velocity ($F(1,127) = 0.02$, $p = 0.90$, ANOVA), or position error ($F(1,127) = 0.002$, $p = 0.97$, ANOVA) (Supplementary Fig 3a–c). As for the delay condition, the animals strive to keep the cursor near the target center and there is no change in strategy. Additionally, performance did not improve over time, with position error magnitude increasing slightly over trials ($\rho = 0.29$, $p = 0.03$) and recording days ($\rho = 0.62$, $p = 0.0002$) in the hidden cursor condition (Supplementary Fig. 3d, e). The absence of adaptation in both delay and hidden condition implies that the internal model, within and across days did not adapt. As for the delay, the hidden cursor condition increases the difficulty of the

task as position error magnitude increases (Supplementary Fig. 3f, $t(35) = -9.12$, $p < 0.001$).

## Discussion

By using two manipulations of visual feedback during pseudo-random tracking, this study reveals key properties of the SS modulation with performance errors and arm kinematics. Delaying the visual feedback shifts the timing of the predictive position error modulation to earlier leads by an interval matching the experimental delay, while preserving the timing of the feedback modulation. This is consistent with a forward internal model that generates predictions with respect to the movement of the hand but not to the delayed movement of the cursor. Conversely, the lagging modulation with position is coupled to the visual feedback. Reducing the visual feedback inside the target decreases the SS feedback modulation with position error inside the target without affecting the predictive modulation. The results from both experiments converge to the same conclusions that the leading representation of position error is time-locked to hand movement and therefore, consistent with a forward model that transforms motor commands into sensory predictions. Furthermore, the invariance of the feedback position error encoding in the delay condition and the decrease in the SS feedback modulation inside the target in the hidden cursor condition show that the lagged SS modulation with position error is driven primarily by visual feedback.

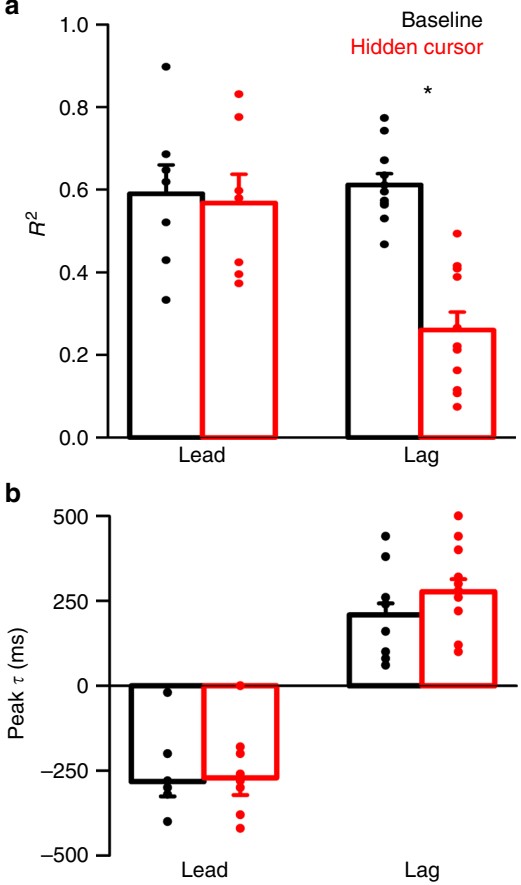

**Fig. 7** Hidden cursor population results for position error encoding. **a** Average encoding strength ($R^2$) of both lead and lag encoding for the 11 Purkinje cells with significant decreases in SS encoding of position error in the hidden cursor condition. Analysis of variance yielded a significant interaction between the feedback condition and the timing of encoding ($F(1,35) = 10.65$, $p = 0.0026$, ANOVA) with a significant decrease in the amplitude of the lagging but not the leading $R^2$. **b** Average timing of lead and lag SS encoding of performance in baseline and hidden cursor conditions is not significantly different ($F(1,45) = 1.04$, $p = 0.31$, ANOVA)

The selective effects of manipulating the visual feedback demonstrate the independence of the position error and kinematic SS signals as well as the independence of the SS predictive and feedback modulation with position error. The independence of the kinematic and error signals supports the concept that two forward models are operative, one for the limb and the other task performance[2]. Purkinje cells have intrinsic firing[28,29] that is modulated by 100,000 to 200,000 parallel fiber-Purkinje cell synapses[30] with less than 200 active synapses required to generate a SS[31]. Therefore, Purkinje cells have the bandwidth to carry a large number of independent signals. Cerebellar granule cells, the origin of parallel fiber inputs to Purkinje cells, are modulated by a host of both sensory and motor parameters[32–35] and acquire predictive signals during learning[36,37]. Therefore, Purkinje cells have the necessary inputs for the SS firing to both predict the consequences of motor commands and signal the corresponding sensory feedback for multiple forward models.

Similar to our previous studies, individual Purkinje cells encode a combination of position error and kinematics[16,20–22]. Further supporting the observation of multiplexing in the cerebellum, SS discharge simultaneously conveys signals on the timing and on-line control of saccadic eye movements[38]. Although

each Purkinje cell encodes a heterogeneous combination of signals, at the population level there is considerable information about the past and future trajectories of each behavioral variable[16,20]. We suggest that this functional heterogeneity provides for a high-fidelity representation of behavior that is flexible enough to compensate for perturbations, as shown in this study by successfully tracking during degraded visual feedback.

The longstanding view is that movement errors are encoded by complex spike discharge[39–42]. The present study extends the hypothesis to involve the SS discharge of Purkinje cells in error signaling, including playing a role in SPEs. In a variety of tasks, SS discharge modulates with errors (for review see ref.[15]), in particular during pseudo-random tracking[16,20,21]. Also, changes in SS discharge are sufficient to drive learning[43,44], including in the absence of climbing fiber input[45]. The finding that Purkinje cell SS encoding of position error responds causally to experimental manipulations shows that SS discharge plays a major role in the cerebellar error processing.

Predicting the consequences of motor commands by a forward internal model has many powerful uses in the nervous system. Predictions allow for state estimation, state prediction, anticipation of future events, and enhanced sensory perception[4,46–48]. Moreover, the predictive signals can be used to cancel self-generated sensory reafferent signals and to determine SPEs[2,4]. The present findings have several implications for the cerebellum functioning as a forward internal model including a role in computing SPEs. The selective effects of the feedback manipulations introduce mismatches between the predictive and feedback SS modulation relative to the baseline condition. The delay and hidden cursor result in a timing and amplitude mismatch, respectively. The discrepancies are accompanied by increases in position error magnitude. Therefore, the feedback driven changes in SS modulation with position error is consistent with the predictive and feedback components of a SPE.

For the cancellation of sensory reafferents and determining SPEs, an outstanding question is where the comparison of predictive and feedback signals occurs. The deep cerebellar nuclei (DCN) are possible candidates for the initial step in this integration due to the convergence of numerous Purkinje cells onto a DCN neuron[49,50]. Responses of nuclear neurons are highly dependent on the synchronicity of Purkinje cell SS firing[49,51,52], and therefore, could integrate the predictive and feedback signals in a population of Purkinje cells to estimate their mismatch. Supporting this concept is the increased modulation of rostral fastigial nucleus by passive rather than self-generated motion[53]. Similarly, dentate neurons appear to encode the omission of an expected stimulus, possibly a correlate of a SPE[54].

While our results focus on prediction and feedback in the motor control domain, recent studies further demonstrate the of prediction errors in a spectrum of behaviors. For example, the primary visual cortex detects mismatches between the actual visual feedback and sensory predictions based on motor and visual inputs[56]. Neurons in the anterior cingulate cortex encode an abstract representation of expected outcomes that switches to a representation of the actual outcome when expectations are not met, pointing to high-level processing of prediction errors[57]. Therefore, the processing of prediction errors takes place in numerous brain structures and is important for learning and controlling behavior[2,55,58,59].

Although the predictive and feedback SS modulations are mismatched and position error increases, the animals can still perform the task during the feedback manipulations. The conservation of predictive SS modulation with errors and kinematics during the feedback manipulations suggests that the internal model is robust and generates accurate predictions based on the motor command and the state of the effector, even with sub-

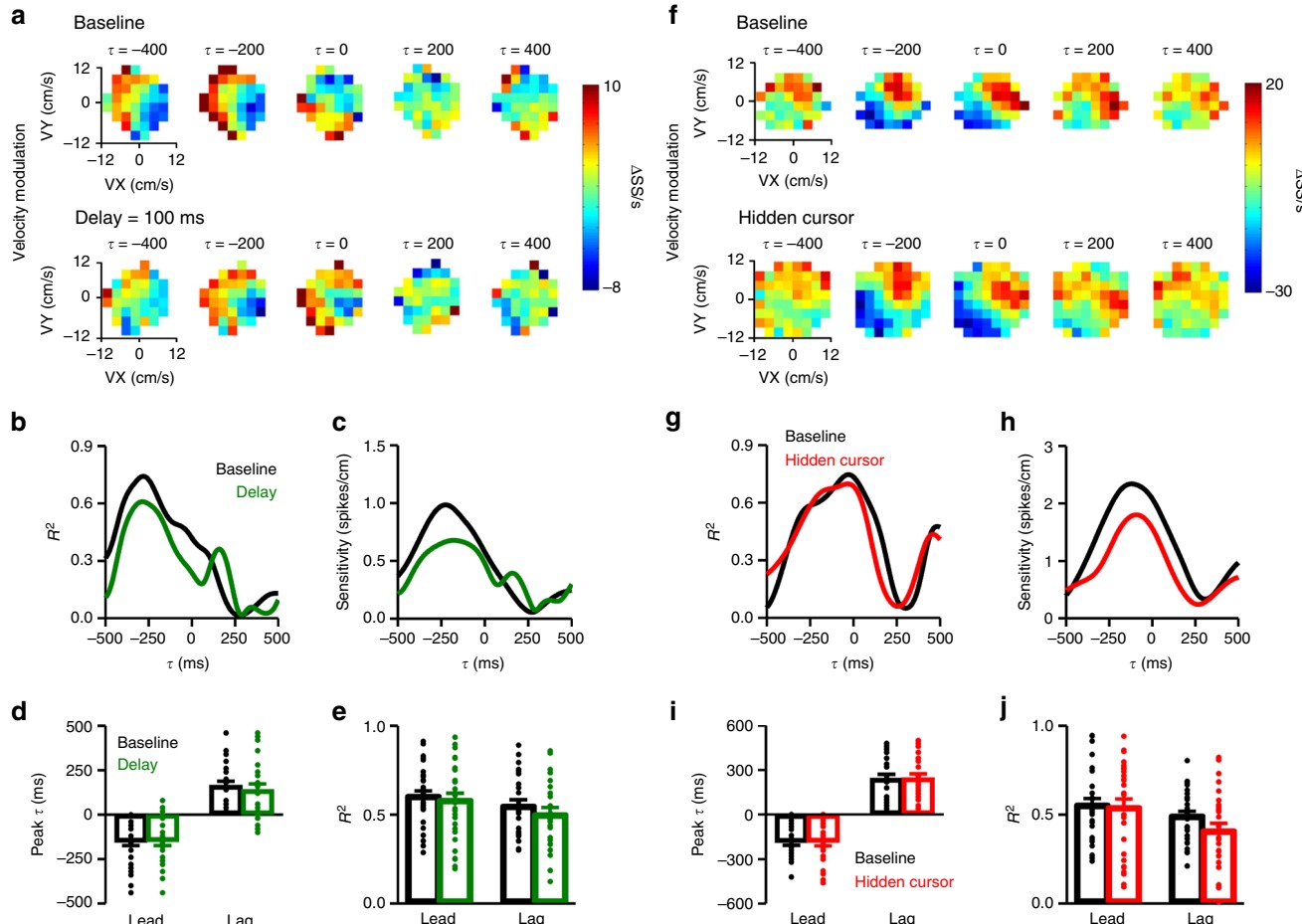

**Fig. 8** Kinematic encoding is unaffected by visual feedback manipulations. **a** Firing maps for an example Purkinje cell with lead SS velocity modulation in both baseline (top row) and 100 ms delay (bottom row) conditions. Each map indicates SS modulation at a specific lead (negative $\tau$) or lag (positive $\tau$). **b** and **c** $R^2$ and sensitivity profiles, respectively, for the example Purkinje cell shown in **a** with significant encoding of manipulandum velocity in both baseline (black line) and 100 ms delay (green line) conditions. **d** Average peak timing of both lead and lag SS encoding of manipulandum kinematics did not change between baseline and delay conditions ($F(1,107) = 0.15$, $p = 0.70$, ANOVA). **e** Average encoding strength ($R^2$) of lead and lag encoding for all Purkinje cells with significant encoding of kinematics ($F(1,107) = 0.47$, $p = 0.71$, ANOVA). **f** Firing maps for an example Purkinje cell with lead SS velocity modulation in both baseline (top row) and hidden (bottom row) conditions. Each map indicates SS modulation at a specific lead (negative $\tau$) or lag (positive $\tau$). **g** and **h** $R^2$ and sensitivity profiles, respectively, for the example Purkinje cell shown in **f** with significant encoding of manipulandum velocity in both baseline (black line) and hidden cursor (red line) conditions. **i** Average peak timing of both lead and lag SS encoding of manipulandum kinematics in baseline and hidden cursor conditions ($F(1,113) = 0.01$, $p = 0.93$, ANOVA). **j** Average encoding strength ($R^2$) of lead and lag encoding for all Purkinje cells with significant encoding of kinematics ($F(1,113) = 1.48$, $p = 0.23$, ANOVA)

optimal visual feedback. As demonstrated previously, the feedforward SS modulation carries the information needed to make accurate predictions about future performance errors and kinematics[16,20]. Psychophysical studies show the CNS employs stable predictions to control movements during distortions of the visual feedback[47]. Presumably, the cerebellum uses the stable representations in the Purkinje cell discharge to facilitate accurate control of even complex tracking behaviors under noisy conditions.

## Methods

**Animal use statement**. All animal experimentation was approved by the Institutional Animal Care and Use Committee of the University of Minnesota and conducted in accordance with the guidelines of the National Institutes of Health. The data that support the findings of this study are available from the corresponding author upon reasonable request.

**Random tracking**. This study utilized two variants of the pseudo-random tracking paradigm described previously[16,22,60]. Two rhesus monkeys (8-year-old female and 15-year-old male) were trained to use a robotic manipulandum (InMotion²,

Watertown MA) that controls a cross-shaped cursor to track a circular shaped target (2.5 or 3.5 cm in diameter) on a computer screen. The paradigm started with an initial hold inside a stationary target for a random period of time (500–3000 ms). The initial target position on the screen was randomized. Next, the target moved along a trajectory selected randomly from 100 trajectories defined a priori. Pseudo-random target paths were generated from a sum of sine waves and ranged from 3–10 s. Target speed was randomly varied in each path so that the average speed was approximately 4 cm/s and conformed to the two-thirds power law[61,62]. The trajectories were low-pass filtered and selected to avoid sharp turns and large changes in speed, and ended with a final hold period of 500–3000 ms. The paradigm required that the monkey maintain the cursor within the target and allowed only brief excursions outside the target (<700 ms), otherwise the trial was aborted. Note that the hold times, length of trajectories, and duration of permitted "excursions" were modified from our original studies to accommodate the increased difficulty of tracking in the altered feedback conditions[16,22]. Similarly, because of the differences in each animals ability to perform the task with the feedback manipulations, one monkey performed all task conditions with a 2.5-cm target diameter and the other with a 3.5-cm target diameter. Pseudo-random tracking has several advantages compared to other tasks including providing more comprehensive and uniform coverage of parameter workspaces and dissociating kinematic from error parameters[22,60].

Hand movements were measured by the transducer on the manipulandum and then mapped onto the computer screen as the cursor movement. Hand (X and Y)

and target position were sampled at 200 Hz. Hand velocity (VX and VY) was derived by numerical differentiation of the hand position. Position error (XE, YE) was defined as the difference between cursor and target positions (Fig. 1d). For further analyses, behavioral parameters were down-sampled to 50 Hz. For analysis of the behavior, we also determined position error magnitude, defined as the absolute value of position error. Because the animals used different sized targets, we normalized position error magnitude to the baseline condition for each monkey (Supplementary Figs. 2d–f and 3d–f).

**Visual feedback manipulations.** This study applied two novel manipulations of visual feedback during pseudo-random tracking to test whether the SS modulation with performance errors and kinematics represents the predictive and feedback components of SPE. For a given recording session, first a block of baseline trials were collected. For the second block, one of two manipulations of the visual feedback were implemented during the actual tracking. In the delay condition, visual feedback was delayed by introducing a 100 or 200 ms lag between manipulandum and cursor movements (Fig. 2). In the hidden cursor condition, the cursor was not displayed when inside the moving target (Fig. 5). Both conditions increased the difficulty of the task, however not to a degree that prevented the animals from performing the required trials. Recording sessions typically consisted of blocks of 50 baseline trials followed by 50 visual feedback manipulation trials, though in many cases more trials were collected. Importantly, visual feedback manipulations were imposed only during the tracking and were removed in the inter-trial intervals, during which the animal moved the cursor to a new target start position, in order to reduce any adaptation effects (see Supplementary Figs. 2 and 3).

**Surgical procedures and data collection.** In each animal, head restraint hardware and a recording chamber were chronically implanted over the parietal cortex ipsilateral to the arm used to track. As described previously[22], electrode targeting and recording locations were determined by a combination of MRI and CT imaging using the software Monkey Cicerone[63] and anatomical modeling. In both animals, modeling of the recording chamber and electrode positions showed that an electrode placed in the center of the chamber would target the intermediate zone close to the primary fissure. In addition, the recording grid was restricted to ~5 mm × 5 mm in the center of the chamber. Therefore, based on the imaging, most Purkinje cells were recorded primarily in lobules IV–VI of the intermediate zone.

After full recovery from chamber implantation surgery, extracellular recordings were obtained during normal daytime hours using Pt–Ir electrodes with parylene C insulation (0.8–1.5 MΩ impedance, Alpha Omega Engineering, Nazareth, Israel). Purkinje cells were identified by the presence of CSs followed by the characteristic pause in SS activity (see Fig. 1b and c) and recorded using previously established methods[17,21]. After conventional amplification and filtering (30 Hz–3 kHz band pass, 60 Hz notch), SSs were discriminated online using the Multiple Spike Detector System (Alpha Omega Engineering, Nazareth, Israel). Resulting spike trains were digitized and stored at 1 kHz. The raw electrophysiological data were also digitized and stored at 32 kHz. Neither the recordings nor analyses were done blinded.

**Analysis of SS modulation.** The goal of the analysis was to determine how the two feedback manipulations changed the SS modulation with position error (XE and YE) as well as the two kinematic variables, hand position (X and Y) and velocity (VX and VY). Typical of electrophysiological studies in behaving monkeys, the neuronal sample size is similar to previous publications studying Purkinje cell SS activity. Using the fractional interval method, the SS trains were transformed to a continuous firing rate in 20 ms bins (Fig. 1e)[64]. The SS firing rates were filtered using a second order lowpass, butterworth filter with a 3 Hz cutoff. For display and analyses, for each trial the mean firing rate was subtracted from the instantaneous firing rate. For each behavioral parameter, the mean-subtracted SS firing rate data from all trials were partitioned and averaged into 64 (8 × 8) equal bins as follows: −2 to 2 cm (0.5 cm × 0.5 cm bin) for XE and YE, −6 to 6 cm (1.5 cm × 1.5 cm bin s) for X and Y, and −12 to 12 cm/s (3 cm/s × 3 cm/s bin) for VX and VY.

SS firing relative to both error and kinematic parameters was analyzed using temporal linear regressions[16,22]. The goal of this initial analysis was to determine the predictive and/or feedback SS modulation with the behaviors of interest: position, velocity, and position error. To determine the temporal relationship between SS discharge and behavior, the firing was shifted relative to each parameter in 20 ms steps, ranging from −500 to 500 ms (negative values representing firing leading behavior). At each step, the correlations between SS firing and position error (XE and YE) and kinematic (X and Y or VX and VY) were assessed and the time of the best fit lead and lag ($\tau$-value) between SS activity and behavioral parameters were determined[16,23–25,60]. Note that use of $\tau$ to denote lead and lag firing with behavioral parameters is equivalent to the $\Delta t$ or $\Delta\delta$ used in other publications[24,25,65].

The correlation between firing and each behavioral parameter was determined in two steps. First, SS variability associated with the rest of the parameters were removed by determining the firing residuals from a multi-linear model of SS firing that includes the kinematic and error parameters except the parameters of interest. Second, the firing residuals were then regressed against the parameters of interest, generating the $R^2$ and regression coefficient ($\beta$) profiles as a function of $\tau$. For

example, the firing residuals needed to evaluate the SS modulation with position error are obtained by regressing actual firing ($F$) to this multi-linear model:

$$F(t+\tau) = \beta_o(\tau) + \beta_X(\tau)X(t) + \beta_Y(\tau)Y(t) \\ + \beta_{VX}(\tau)VX(t) + \beta_{VY}(\tau)VY(t) + FR(t+\tau). \tag{1}$$

The resulting firing residuals (FR) were then regressed to the two position error terms, XE and YE:

$$FR(t+\tau) = \beta_0(\tau) + \beta_{XE}(\tau)XE(t) + \beta_{YE}(\tau)YE(t) + \varepsilon(t). \tag{2}$$

For each $\tau$, this regression results in an $R^2$ value indicating goodness of fit of the SS firing to both XE and YE, and two regression coefficients ($\beta$), one for each error parameter. The overall SS sensitivity to position error were computed from the two regression coefficients:

$$\text{Sensitivity} = \sqrt{\beta_{XE}^2 + \beta_{YE}^2}. \tag{3}$$

The significance of the $R^2$ at each $\tau$-value was assessed against a noise distribution of shuffled data. $R^2$ values were obtained from 100 repeats of the same regression analysis performed on firing and behavioral data uncoupled through random trial shuffling. The threshold for significance was defined as the mean ± 3 SD of the shuffled distribution. For each parameter, significant correlations were defined if a local maximum of the $R^2$ profile at either predictive or feedback timings exceeded the trial shuffled noise level. Then, the timing ($\tau$-value) of the peak lead and/or lag was determined. A similar analysis as outlined in Eq. 1–3 was undertaken for velocity and position.

**Analysis of visual feedback delay.** For the visual feedback delay paradigm, our primary hypothesis was that introducing a lag between the movement of the manipulandum and the movement of the cursor would affect the timing of the SS encoding of position errors. Therefore, we determined the peak predictive and/or feedback timing ($\tau$) of the SS modulation with each behavior of interest under both baseline and delay conditions using the regression analysis described above. To ensure that we were accurately comparing the same SS signals between baseline and delay, peaks were selected for analysis if they were present in both baseline and delay conditions with the same sign regression coefficients (e.g., positive modulation with XE and YE in both baseline and delay). The $R^2$ value at the peak predictive and feedback times was also determined and compared for baseline and delay conditions.

**Analysis of visual feedback reduction.** During the hidden cursor, the effect of hiding the cursor while it is inside the target creates two conditions: one where there is no visual feedback available (cursor inside target boundary) and one where there is visual feedback available (cursor outside target boundary). Thus, to assess the effects of the hidden cursor paradigm, we performed a similar temporal linear regression analyses (Eqs. 1–3), but only using the data points recorded when the cursor was inside the target. The resulting $R^2$ and sensitivity profiles provided a quantification of the SS encoding of behavior inside the target, both predictive and feedback, for the baseline and hidden cursor conditions. Encoding decreases were considered significant if the peaks exceeded the statistical threshold for significance in the baseline but not hidden cursor conditions.

**Analysis of kinematic modulation.** We also assessed the effects of visual feedback manipulation on the SS encoding of kinematics. For the delay condition, we determined the timing of SS modulation with hand kinematics (position and velocity) in both baseline and delay conditions using the same methods as those described for position error. For the visual feedback reduction, we determined the magnitude of predictive and feedback modulation with kinematics in both baseline and hidden cursor conditions. Importantly, we restricted the analysis to the behavior occurring only when the cursor was inside the target center as for position error.

**Code availability.** All analyses were performed using custom MATLAB code, which will be made available upon request to the authors.

**Data availability.** All data are available on request to the authors.

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

## Acknowledgements

We wish to thank Dr. Claudia Hendrix and Lijuan Zhou for technical support. Supported in part by NIH grants R01 NS18338 (TE), T32 GM008471(TE), and F31 NS095408 (MS) and NSF grant IGERT DGE-1069104 (MS).

## Author contributions

T.E., M.S., and L.P. designed the paradigm and analyses. M.S. collected and analyzed the data with input from T.E. M.S., L.P., and T.E. wrote the manuscript.

## Additional information

**Competing interests:** The authors declare no competing interests.

