## [Peer Review File(PDF 350 kb) · Nature Communications]

Reviewers' comments:

Reviewer #1 (Remarks to the Author):

This paper extends an analysis that the authors have been publishing in a number of nice papers for several years. They had formed the hypothesis based on a correlational analysis that cerebellar Purkinje cells represent errors in tracking arm movements twice – first as a prediction of the error based on a forward model that converts corollary discharge of the motor commands into predictions of the forthcoming sensory errors, and second as the actual error based on sensory feedback. In the present paper, they move partway from correlational evidence to causal evidence by experimental manipulations that alter the two representations separately and independently, and in the way predicted by their hypothesis. I think this is the real strength of the paper and, for me, it provided the first demonstration of “what an internal model would look like in neural discharge”, and so I think it is an important advance.

While I think the results of the paper are important, I do have a number of fairly serious concerns. I think that most of my concerns can be addressed by the authors, but I want to be clear that a cosmetic revision of the current paper will not suffice. No additional work is needed, just better communication of the work.

The paper is challenging to read and understand. Indeed, it is hard to find a clear statement about the main conclusions in the right places. I think this paper shows that (1) the leading representation of error is time-locked to the movement and therefore is the output of an internal model that converts corollary discharge into sensory predictions and (2) the lagging representation is driven directly by sensory inputs. I had to study the paper quite hard to figure out each piece of it, even though I am (1) directly in this field and (2) fairly quantitative and computationally minded. My fear is that the paper, as written, will be understood and appreciated by very few readers. This needs to be corrected.

I realize that much of the analysis method has been used in prior papers, but I didn't fully understand it in those papers either. Because I was reviewing this one, I had to fight my way through it. As few suggestions:

(1) It would be very helpful to have a clear statement of how the correlation analysis was done. If I understand correctly, the authors broke the data into time bins. They measured the tracking error in each bin and then asked how the simple spike firing rate at different values of Δt relative to the time of the measurement was correlated with the tracking error. They did the same analysis for kinematics, except that the primary measurement was of a kinematic variable. Crazy to say, but there is not simple statement of how they did this in the paper.

(2) Overall, I find that the presentation of exactly what was done is a little cursory. The Methods gives a broad description, but many important minor details are missing. I would ask that the authors revisit their Methods section and present what they did, especially with the data analysis, more thoroughly. For example, in analyzing simple-spike firing rate, did they subtract “baseline”? Where did baseline come from? The start of each trial? A few trials at the start of the session? These are really important details.

(3) The paper mainly presents reduced data with only the tiniest bit of raw data in the supplementary figures. I think it would be helpful to the reader to show more raw data in the main figures of the paper, and to help the reader to understand how the raw data lead to the quantitative results presented in the current figures. I also would like to see some documentation of how well individual

Purkinje cells show the nice clean results in Figure 3b and c.

(4) For me, “tau” has a very different meaning from “delta-t” and the use of “tau” instead of “delta-t” in the equations, text, and figures was incredibly misleading. I see that they did the same thing in prior papers, but I would say that I would strongly prefer to have this paper easily read and understood, even if it means a lack of consistency of terminology with the prior papers.

(5) Figure 1 is at the same time essential and inscrutable. First, the combination of green versus black and dashed versus continuous curves was difficult to see visually. Second, the text surrounding the figure, including the figure legend, was nearly inscrutable. For my first reading, the only thing I could do was to decide to trust the authors, but I did not understand this important set-up figure until I came back and studied it. I think that part of the confusion was that I wasn’t clear to what the measurements were referenced. When the feedback is delayed by 200 ms, are the referenced to the error of the hand position, or the error of the delayed cursor position? Now I understand and I get it, but this simple point was a point of major confusions.

The paper never makes a clear statement that a 200 ms time advance of the correlations means that the simple-spike firing was time-locked to the movement itself, rather to the feedback from the movement. This is the critically important piece of the presentation because it means that this leading component of the simple-spike firing was based on signals related to execution of the movement, but was related to a prediction of the upcoming error. Therefore it is the neural representation of an internal model. Critically important. This is one of the main conclusions of the paper, but the authors do not write clearly about it. It was much easier to understand that the disappearance of the lagged relation to error in the hidden-cursor task meant that that component of the simple spike firing was driven by sensory feedback.

I have concerns about some of the data presentation.

(1) Figures 2 and 4 make the important point that the behavior is fairly stationary. But it does not make sense in terms of communication to put these rather mundane figures in front of the very important Figures 3 and 5. I would like to see the analysis of the behavior put at the end of the Results or even in the Supplementary Material so that the key results can be digested up front.

(2) I appreciate that the data for each cell appear in the summary graphs of Figures 3 and 5 but I miss them in Figures 6 and 7. Frankly, the effect in Figure 6 is not very convincing on the basis of the averages, and I doubt that presenting the individual neurons is going to make it look better. Also, I would say that the point made by Figure 6 is sort of peripheral to the main points of the paper and it could be omitted. Figure 7 seems important, but I also would like to see the individual neurons in the summary graph; .

(3) I am confused about the number of points in Figure 5e. Why are there different numbers of red and black symbols for the lead component?

The Abstract is not very helpful and appears to have been hastily written. It does not flow logically, important points are not defined, and it is not useful for a reader who hasn’t already read the full paper.

Reviewer #2 (Remarks to the Author):

The experiments presented in this paper apply visual stimulus manipulations in a tracking task to segregate feedforward and feedback signals produced by cerebellar PCs. There are interesting results which build upon previous studies by the same group.

The manipulations are nicely executed and indicate that leading correlations between simple spike rate and position error exhibit properties of a forward model, whereas lagging correlations encode sensory feedback. My concern is that the study lacks novelty as it mainly confirms the findings from a previous study – Popa et al, J Neuroscience 2012. For this reason, I believe the current paper lacks sufficient impact for Nature Communications.

There are also issues with the way data is presented and analysed that reduce comprehensibility for specialists and non-specialists alike. These are outlined below:

1) The results cannot be appreciated without close attention to previous papers. For example, the temporal linear regression analysis is essential to appreciate the data but is only treated in a cursory manner in this manuscript. Consequently, this does not feel like a standalone study.

- Examples of assumed knowledge are present in the abstract – e.g. it should be noted in the first few sentences that individual PCs display both leading and lagging representations of movement error, with some explanation as to the significance of this arrangement.

2) The clarity of Figure 1 would be improved by adding the behavioural cartoons in Supplementary Figures 1 & 2. Also it would be useful to include an example PC recording demonstrating the basic phenomenon that is being investigated, i.e. a cell that shows both leading and lagging representations of prediction error.

3) Figure 2 is not particularly informative and seems potentially contradictory. Supplementary 1c shows a significant increase in normalized error (using t-test), but Figure 2c and accompanying statistics show no detectable change (using ANOVA). What is going on? Is the large number of zero values in the ANOVA diluting significance? Potentially the 1cm resolution of X and Y error is insufficient. Have the authors tried looking at X/Y errors with finer resolution?

4) A similar issue exists between Figure 4 and Supplementary Figure 2. In addition, the units for reporting error are inconsistent in Supplementary Figure 2c (reported in centimeters).

5) Bar plots are used in Figures 3&5 whereas whisker/box plots are used in Figures 6&7. What is the reason for this inconsistency?

6) In general, the conceptual importance of the results presented in Figure 6 and 7 is unclear. This particularly applies to Figure 7 – what is the interpretation/significance of this result? At present these data feel supplemental in nature.

7) Do the authors have any explanation for why the error increases over time/trials for the visual feedback reduction experiments. Is there a correlate of this increase in the lagging PC simple spike response?

8) Is the increased SS feedback modulation outside the target due to increased visual stimulation (relative to inside the target). Can the authors find evidence to rule this out?

9) The cell count reporting for the visual delay experiments are not transparent. They should be reported in the same way as for visual feedback removal.

Reviewer #3 (Remarks to the Author):

This study describes experiments designed to experimentally test the hypothesis that the cerebellum implements an internal model, that predicts the sensory consequences of motor commands by computing a sensory prediction error. To do this the authors implement a clever experimental design involving recording simple spike activity in Macaque cerebellum during a pseudo-random tracking task that also includes visual occlusion. This enable them to separate sensory prediction from visual feedback. The results show that leading temporal components of simple spike activity encode sensory prediction error, while components that lag the target contain information visual feedback. Importantly they show that these Purkinje cell responses encode position error rather than kinematic properties, such as velocity. The data and analysis are of high quality and the main results are sufficiently important, to be of wide interest to the community. But, unfortunately, the current presentation of the study is poor, making the study inaccessible. Indeed, as a cerebellar physiologist, I found it difficult to read, so a non-specialist would find it impenetrable. Some simple modifications, defining terms in the text rather than the methods, explaining concepts in clear language, presenting a cartoon of the task and the raw data in the first figure of the main manuscript with an illustration of how it is analysed, would help a lot. However, I think that more is needed for a Nature Communications audience, so I would recommend a major rewrite, incorporating feedback from colleagues who are not in this field.

Specific points

- 1) Figure 1 should provide an illustration of the setup and task, together with raw simple spike firing data, its modulation, how it is analysed in terms of lead and lag, as well as an illustration of the positional error. Panels a and b in Supplementary figure 1 could be used as a basis.
- 2) Define all parameters that are used in the text and figures as they arise in the main text. These include Position error $XE, YE, R2, \tau, D\tau$, etc. For several of these the definition in the Methods could be used.
- 3) Given that the analysis in figure 7 is to control for encoding of kinematics versus prediction error, it seems more logical to refer to the figure early in the text, thereby strengthening the interpretation of the analysis. It could be made a supplementary figure to prevent disruption of the narrative flow.
- 4) It is not clear why Purkinje cell recordings were made from such a large area (lobules IV-VI). The authors need to be more specific in term recordings location (Zone, zebrin II band) as studies suggest that cerebellar function varies with location. Was there any difference between lobules? Between vermis and paravermis?
- 5) In relation to point 4, it is clear that the functional properties of Purkinje cells is highly heterogeneous. While this is reported, the authors need to do a better job in describing the criteria they have used to classify responsiveness and it would be instructive to have a visual presentation of the different classes of cells and their frequency. Lastly, they should clearly identify where in the distribution the cells used to illustrate the prediction error and feedback error are found.
- 6) The firing maps are difficult to read, as there are several variables. Is it possible to simplify the

presentation so that only the variables of interest are presented? For example, does the direction of the simple spike modulation matter? Could a distance from target center representation be used?

7) In general, the figures require a clearer explanation for the people not familiar with these types of behavioral experiments.

a. Is the target a big red circle as in figure 1C? If the grey zones in figure S1a and S2a correspond to the moving target, could you make the color match?

b. Does the black cross in figure 1C corresponds to what the monkey sees when he moves the manipulandum?

c. Why do trials have different durations? Are the trials interrupted if the monkey stay out of the area for >700ms?

d. Why do cursors have different size? Is it an adjustment specific to the monkey or to the experiment? Is there any protocol for this adjustment?

8) P4 : removing the tracking cursor is described as "reduction of visual feedback" while it is actually more a suppression of all informative visual feedback

9) P6 and elsewhere there are statements in the text that should have a figure panel citation.

10) P6 "the average encoding strength of position error is not affected by delay" seems contradictory with Fig S1c

11) P9 "visual movement of the cursor" is rather vague, and could relate to eye movements or sensory processing of the target. Please be more specific.

12) Discussion. The discussion of the number synapses required to generate a simple spike is rather outdated, given that Purkinje cells are intrinsically active even in the absence of inputs in the slice preparation.

13) Discussion p10, In that context, do the authors means "Purkinje cell have the capacity to receive..."

14) Discussion. It would be useful to set the current results in the context of recent work in other brain areas showing that expectation mismatch signals are widespread (e.g. Work from Georg Keller in visual cortex). Indeed, could some of the current results, such as the enhanced feedback responses outside the target, be due to the system responding more strongly to unexpected events?

15) P12. Should "omission of unexpected" read "omission of expected"?

16) Figure 2d,e Green symbols cannot be distinguished from black.

17) Figure 7. Given basic plots used are similar to those used previously for error, annotation indicating data for velocity are being presented would be useful.

18) Although it is stated that only lag difference is statistically significant, (Fig6A and 6B & page 8), the figure used for illustrating the lead condition and the much larger error suggest that the lead condition is also affected

19) Complex spikes are expected to contribute to error encoding. Although they are mentioned in the discussion, they are completely ignored from the paper. It would be relevant to see how they fire

compared to simple spike modulation or, at least, what is their contribution is to simple spike firing as they are likely to introduce pauses.

Revision of NCOMMS-17-21335-T

Response to Reviewers:

We wish to thank the Reviewers for their insightful and helpful comments. We have made every effort to respond to each concern and question. A major concern of all three Reviewers was that the presentation lacked clarity and was not accessible to a general readership. Therefore, we have undertaken a major revision that includes substantial reorganization, modifying as well as adding new figures and clarifying the methods. The main conclusions have been clearly stated. In our opinion, the changes have resulted in a substantially improved manuscript. The following are point-by-point responses to the Reviewers' comments.

Reviewer #1 (Remarks to the Author):

First, we would like to thank the Reviewer for the supportive comments stating that the manuscript provided the first demonstration of "what an internal model would look like in neural discharge", and so I think it is an important advance. "

The paper is challenging to read and understand. Indeed, it is hard to find a clear statement about the main conclusions in the right places. I think this paper shows that (1) the leading representation of error is time-locked to the movement and therefore is the output of an internal model that converts corollary discharge into sensory predictions and (2) the lagging representation is driven directly by sensory inputs. I had to study the paper quite hard to figure out each piece of it, even though I am (1) directly in this field and (2) fairly quantitative and computationally minded. My fear is that the paper, as written, will be understood and appreciated by very few readers. This needs to be corrected.

I realize that much of the analysis method has been used in prior papers, but I didn't fully understand it in those papers either. Because I was reviewing this one, I had to fight my way through it. As few suggestions:

In response to the Reviewer's general concerns, the revision strives to increase the readability and make the study more accessible to general readers. We added two new Figures (2 and 5) that outline the rationale for the delay and hidden cursor condition and the expectations for the changes in the simple spike firing in the context of a forward internal model. In response to Reviewer 3, a new Figure 1 provides an overview of the methods, example recordings and the analysis approach. Throughout the Results we emphasize the link between the findings and the forward model hypothesis. We also reworked the Materials and Methods to provide the critical details as well as expand on the analyses, again with the goal of increasing readability.

Concerning the need for a clear statement about the main conclusions, we made changes to the Abstract, Results and Discussion that clarify the key findings. For example the Abstract now states: "Consistent with the neural representation of a forward model, delaying the visual feedback demonstrates that the simple spike prediction of position error is time-locked to the hand movement. Reducing the feedback shows that the lagged modulation is directly driven by visual inputs. Therefore,

Purkinje cell discharge carries both predictions based on motor commands and the sensory feedback necessary to compute sensory prediction errors.” In addition, the changes to Figures 2 and 5 illustrate these concepts.

(1) It would be very helpful to have a clear statement of how the correlation analysis was done. If I understand correctly, the authors broke the data into time bins. They measured the tracking error in each bin and then asked how the simple spike firing rate at different values of delta-t relative to the time of the measurement was correlated with the tracking error. They did the same analysis for kinematics, except that the primary measurement was of a kinematic variable. Crazy to say, but there is not simple statement of how they did this in the paper.

Both in the revised Results and in the Materials and Methods, we reworked the description of the linear regression analysis. Before detailing the linear regression analysis, we added an overview of how we determine the lead and lag simple spike firing relative to the behavior and that the analysis is done in 20 msec steps for the parameters analyzed (position error, position and velocity). As noted above, Figure 1 also introduces these concepts.

(2) Overall, I find that the presentation of exactly what was done is a little cursory. The Methods gives a broad description, but many important minor details are missing. I would ask that the authors revisit their Methods section and present what they did, especially with the data analysis, more thoroughly. For example, in analyzing simple-spike firing rate, did they subtract “baseline”? Where did baseline come from? The start of each trial? A few trials at the start of the session? These are really important details.

We revised the Materials and Methods, particularly with respect to the analyses with the goal of making certain that important details are not missing as well as not requiring a reader to rely on our previous publications. This includes, as suggested by Reviewer 3, that we provide in Figure 1 an overview of the methods, recordings, and analyses. For the specific question on the simple spike firing rate, we clarify in the Materials and Methods that “For display and analyses, the mean firing rate for each trial was subtracted from the instantaneous firing rate.”

(3) The paper mainly presents reduced data with only the tiniest bit of raw data in the supplementary figures. I think it would be helpful to the reader to show more raw data in the main figures of the paper, and to help the reader to understand how the raw data lead to the quantitative results presented in the current figures. I also would like to see some documentation of how well individual Purkinje cells show the nice clean results in Figure 3b and c.

We include in the Results an overview of the analyses to assist in understanding how the data was processed and analyzed. Regarding the raw data and given the comments of Reviewer 3, we moved the two examples of Purkinje cell recordings from the original Supplementary material to the new Figure 1. In addition, Figure 1 diagrams the concept of lead and lag simple spike firing relative to position error and the firing’s relation to the regression results (Fig. 1d).

Given the pseudo-random nature of the task and the behavior with different trajectories presented in each trial and the variability in Purkinje cell simple spike firing, it is not possible to show the simple spike

firing on a single trial and make meaningful claims about the relationships with the behavioral parameters or how those relationships change with one of the feedback manipulations. To rigorously quantify the simple spike firing relative to the behavior requires a global evaluation of the firing across all the trials to reveal significant relationships.

However, we would stress that the simple spike firing maps are relatively unprocessed data, as these plots are the average, mean-subtracted simple spike firing rate partitioned into 64 (8 x 8) bins at lead and lag times (τ -values) relative to a behavioral parameter. These firing plots provide a visual depiction of the simple spike modulation with position error (Figs. 3 and 6) or the kinematic (Fig. 8) workspaces.

Concerning “documentation of how well individual Purkinje cells show the nice clean results in Figure 3b and c”, we would stress that Figure 3b and c are the results from the example Purkinje cell shown in Figure 3a. The population data in Figure 4 show the results for all 30 Purkinje cells modulated with position error in both the baseline and delay conditions, including the variability around the mean. Figure 4b plots the time shift for each Purkinje cell in the delay condition. The population results show the clear shift in the predictive modulation to more negative τ -values for both the 100 and 200 msec delay conditions and that the timing of the feedback modulation does not change (Fig. 3a-c). Furthermore, we show that the delay condition did not change the amplitude of the lead or lag R^2 (Fig. 3d).

(4) For me, “tau” has a very different meaning from “delta-t” and the use of “tau” instead of “delta-t” in the equations, text, and figures was incredibly misleading. I see that they did the same thing in prior papers, but I would say that I would strongly prefer to have this paper easily read and understood, even if it means a lack of consistency of terminology with the prior papers.

In reviewing the literature, there appears to be no standard concerning the use of “ τ ” or “ Δt ”, to denote lead and lagging firing relative to behavior for the temporal regression analysis used in this study. A series of previous papers (not just our publications) used τ , including the original paper on random tracking¹⁻³. Other studies used Δt ^{4,5}, while others used “ $\Delta\delta$ ”^{6,7}. Given the lack of a standard and that many authors have used “ τ ” we have elected to keep “ τ ”, which also maintains consistency with our previous publications. However, we note in the description of temporal linear regressions in the Materials and Methods that “ τ ” is equivalent to “ Δt ” and “ δ ” used in other publications.

(5) Figure 1 is at the same time essential and inscrutable. First, the combination of green versus black and dashed versus continuous curves was difficult to see visually. Second, the text surrounding the figure, including the figure legend, was nearly inscrutable. For my first reading, the only thing I could do was to decide to trust the authors, but I did not understand this important set-up figure until I came back and studied it. I think that part of the confusion was that I wasn’t clear to what the measurements were referenced. When the feedback is delayed by 200 ms, are the referenced to the error of the hand position, or the error of the delayed cursor position? Now I understand and I get it, but this simple point was a point of major confusions.

Based on these comments and those of the other Reviewers, we created new figures to illustrate the expected results in the delay (Fig. 2) and hidden (Fig. 5) conditions. In the delay condition illustration (Fig. 2), we show a schematic of the lead and lag firing in relation to the hand and cursor movement and how that lead and lag firing is expected to change when the cursor is delayed. It emphasizes that firing modulation is relative to position error, defined as the difference between cursor position and the center of the target. Similarly, Figure 5 clarifies the hidden cursor condition and the expected changes in the lead and lag firing, again based on the simple spike firing relative to position error.

We also stress in the text and in Figure 2 that the simple spike firing modulation with position error is determined relative to the visual feedback on the screen. Conversely, simple spike modulation with kinematics is determined relative to hand movement.

(6) The paper never makes a clear statement that a 200 ms time advance of the correlations means that the simple-spike firing was time-locked to the movement itself, rather than the feedback from the movement. This is the critically important piece of the presentation because it means that this leading component of the simple-spike firing was based on signals related to execution of the movement, but was related to a prediction of the upcoming error. Therefore, it is the neural representation of an internal model. Critically important. This is one of the main conclusions of the paper, but the authors do not write clearly about it. It was much easier to understand that the disappearance of the lagged relation to error in the hidden-cursor task meant that that component of the simple spike firing was driven by sensory feedback.

As noted above, we revised both the Abstract and in the main text (Results and Discussion) to emphasize that during the delayed condition that the predictive simple spike firing with position error is locked to the hand movement and not to the visual feedback. Reducing the feedback shows that the lagged modulation is directly driven by sensory feedback. Therefore, the feedback manipulations demonstrate that the lead simple spike firing with position error is a neural representation of an internal model.

I have concerns about some of the data presentation.

(1) Figures 2 and 4 make the important point that the behavior is fairly stationary. But it does not make sense in terms of communication to put these rather mundane figures in front of the very important Figures 3 and 5. I would like to see the analysis of the behavior put at the end of the Results or even in the Supplementary Material so that the key results can be digested up front.

As requested, we moved the behavioral data to the last section in the Results and moved the behavioral figures into the Supplementary Material (Supplementary Figs. 2 and 3).

(2) I appreciate that the data for each cell appear in the summary graphs of Figures 3 and 5 but I miss them in Figures 6 and 7. Frankly, the effect in Figure 6 is not very convincing on the basis of the averages, and I doubt that presenting the individual neurons is going to make it look better. Also, I would say that the point made by Figure 6 is sort of peripheral to the main points of the paper and it could be omitted. Figure 7 seems important, but I also would like to see the individual neurons in the summary graph;

As requested by both this Reviewer and Reviewer 2, we removed the analyses and the original Figure 6 related to the simple spike modulation outside of the target in the hidden cursor condition. We agree that the results of the simple spike firing with kinematics are important and are retained in Figure 8. These analyses demonstrate that simple spike modulation with kinematics during random tracking reflects the encoding of limb movements, not the feedback associated with movement of the cursor. Further, the observations demonstrate that the simple spike position and velocity representations are independent of position error encoding.

(3) I am confused about the number of points in Figure 5e. Why are there different numbers of red and black symbols for the lead component?

We carefully checked all the points in Figure 5e (now Fig. 6b). For the R^2 plots in “a” and peak τ -value plots in “b”, there are 7 lead data points and 11 feedback data points in both the baseline (black) and hidden cursor (red) conditions. The number of data points can be difficult to discern as individual points can have the same or nearly identical values. For the lead component of the peak τ -value in the baseline condition that the Reviewer has questioned, two cells have peak τ -values of -420msec (note that peak τ -values are computed with a 20 msec resolution). Similarly, for the lag peak τ -values 3 cells have a peak lag of 100 msec, and 2 cells have a peak lag of 300 msec. Therefore, there are only 8 and 9 visually distinct points, respectively. There is a similar overlap with the R^2 values (though obviously the R^2 -values are not exactly the same). For example, one cell in the baseline has a lag peak of 0.5675 and one has a feedback peak of 0.5631, resulting in only 10 visually distinct points.

The Abstract is not very helpful and appears to have been hastily written. It does not flow logically, important points are not defined, and it is not useful for a reader who hasn't already read the full paper.

We extensively revised the Abstract for clarity as well as emphasizing the points raised above by the Reviewer and Reviewer 2.

Reviewer #2 (Remarks to the Author):

The experiments presented in this paper apply visual stimulus manipulations in a tracking task to segregate feedforward and feedback signals produced by cerebellar PCs. There are interesting results which build upon previous studies by the same group.

The manipulations are nicely executed and indicate that leading correlations between simple spike rate and position error exhibit properties of a forward model, whereas lagging correlations encode sensory feedback. My concern is that the study lacks novelty as it mainly confirms the findings from a previous study – Popa et al, J Neuroscience 2012. For this reason, I believe the current paper lacks sufficient impact for Nature Communications.

We thank the Reviewer for acknowledging that the results are of interest and well executed. Concerning novelty, we believe this study does considerably more than confirm the findings from our previous study⁸. Using correlational analyses, the previous study established the presence of position

error signals in the simple spike discharge but lacked causal evidence about the nature of the lead and lag modulation. Using the feedback manipulations, this study provides rigorous evidence that simple spikes encode a prediction of upcoming position error based on the motor commands and sensory feedback of position errors. This study moves our original observations from correlational to causal.

There are also issues with the way data is presented and analyzed that reduce comprehensibility for specialists and non-specialists alike. These are outlined below:

1) The results cannot be appreciated without close attention to previous papers. For example, the temporal linear regression analysis is essential to appreciate the data but is only treated in a cursory manner in this manuscript. Consequently, this does not feel like a standalone study.

We extensively revised the manuscript to minimize relying on previous papers. This includes more details in the Materials and Methods, new figures and schematics that emphasize key concepts and additional explanations in the Results.

- Examples of assumed knowledge are present in the abstract – e.g. it should be noted in the first few sentences that individual PCs display both leading and lagging representations of movement error, with some explanation as to the significance of this arrangement.

We revised the Abstract including in the first few sentences that individual PCs display both leading and lagging representations of movement error.

2) The clarity of Figure 1 would be improved by adding the behavioural cartoons in Supplementary Figures 1 & 2. Also it would be useful to include an example PC recording demonstrating the basic phenomenon that is being investigated, i.e. a cell that shows both leading and lagging representations of prediction error.

In response to this suggestion as well as comments from Reviewers 1 and 3, we added a new Figure 1 that includes a description of the task, examples of single cell recordings in both baseline and feedback manipulations, parameters analyzed, a schematic of the lead and lag representations and the output of the temporal linear regression analysis. As discussed in our response to Reviewer 1, given the pseudo-random nature of the task and the behavior with different trajectories presented in each trial and the variability in Purkinje cell simple spike firing, it is not possible to show the simple spike firing on a single trial and make meaningful claims about the relationships with the behavioral parameters or how those relationships change with one of the feedback manipulations. To rigorously quantify the simple spike firing relative to the behavior requires a global evaluation of the firing across all the trials to reveal significant relationships. However, simple spike firing maps are relatively unprocessed data that provide a visual depiction of the simple spike modulation with either position error (Figs. 3 and 6) or the kinematic (Fig. 8) workspaces.

3) Figure 2 is not particularly informative and seems potentially contradictory. Supplementary 1c shows a significant increase in normalized error (using a Student's t-test), but Figure 2c and accompanying

statistics show no detectable change (using ANOVA). What is going on? Is the large number of zero values in the ANOVA diluting significance? Potentially the 1cm resolution of X and Y error is insufficient. Have the authors tried looking at X/Y errors with finer resolution?

The Reviewer asks why the ANOVA based analysis of the probability density plots of position error in the baseline and delay conditions did not reveal any significance difference (previously Figure 2) versus the significant difference obtained with a Student's t-test performed on position error magnitude (previously Supplementary 1c). The difference is due to the additional averaging and collapsing the data from the 2 dimensional, 8 x 8 probability density plots to the 1 dimensional position error magnitude. Collapsing to position error magnitude decreases the variability and reaches statistical significance between the baseline and feedback conditions as shown in Supplementary Figures 2f and 3f). We have clarified the differences in these behavioral measures in the Methods and Materials.

The Reviewer suggests that the difference could be due to a "large number of zero values", however, this is not the case. Please appreciate, the probability density plots do not have any zero values nor do they have zero variability, instead the bins simply have low probability values (the blue colors).

Finally, the Reviewer requested that we examine the density plots using a higher resolution. Therefore, we increased the behavioral resolution of the probability density plots from the original 8 x 8 binning to 20 x 20 binning (e.g., for position error 0.5 cm x 0.5 cm bins to 0.2 cm x 0.2 cm bins), followed by the ANOVA testing. The higher resolution resulted in a significant effect only when comparing the baseline and the 100 msec delay condition ($F(1, 399) = 3.06, p = 0.036$). All other comparisons were not-statistically significant. Therefore, using finer behavioral resolutions has only a modest effect on the significance testing. As described above, averaging and collapse of the position error from 2 dimensional, 8 x 8 probability density plots to position error magnitude accounts for the difference.

4) A similar issue exists between Figure 4 and Supplementary Figure 2. In addition, the units for reporting error are inconsistent in Supplementary Figure 2c (reported in centimeters).

The Reviewer raises the same concerns for the ANOVA-based evaluation of position error using the 2-dimensional probability density functions of position error in the baseline and hidden conditions versus the significant result obtained with the Student's t-test performed on the position error magnitude. Again, the main difference in the two results is due to the collapsing and averaging position error to position error magnitude. There are no zero-values in the density plots. Also, increasing the spatial resolution did not alter the ANOVA results when comparing baseline versus hidden cursor.

The units for error magnitude in Supplementary Figures 2f and 3f have been corrected.

5) Bar plots are used in Figures 3&5 whereas whisker/box plots are used in Figures 6&7. What is the reason for this inconsistency?

We changed the summary plots to be bar plots throughout the manuscript and all bar plots include the individual data points.

6) In general, the conceptual importance of the results presented in Figure 6 and 7 is unclear. This particularly applies to Figure 7 – what is the interpretation/significance of this result? At present these data feel supplemental in nature.

Reviewer 1 also raised this concern. The original Figure 6 and related results have been removed from the manuscript.

7) Do the authors have any explanation for why the error increases over time/trials for the visual feedback reduction experiments? Is there a correlate of this increase in the lagging PC simple spike response?

We do not have an explanation for the increase in error magnitude over recording days in the hidden condition. However, the magnitude of the change days was modest, approximately a 14% increase. Concerning the question of the relation of the lagging simple spike firing and the error increase, we cannot provide a direct answer as a different Purkinje cell is recorded on each experimental day and the cells have different lead/lag encoding properties. However, we do not believe that the increase in errors has any relation to the lagging responses in the simple spike firing. We observed lead and lagging simple firing in this task and other tasks beginning with the first publications in 2011 and 2012^{8,9} as well as in subsequent publications¹⁰⁻¹².

8) Is the increased SS feedback modulation outside the target due to increased visual stimulation (relative to inside the target). Can the authors find evidence to rule this out?

We cannot completely rule out that in the hidden condition that increased simple spike response outside the target is due to a relative increase in visual stimulation. However, there is no absolute increase in visual stimulation between baseline and the hidden conditions. More likely the increase is due to the increased salience of the cursor outside of the target, as in the hidden condition it provides the only performance feedback to the animal. However, as requested this reviewer and Reviewer 1, we removed the analyses and the original Figure 6 related to the simple spike modulation outside of the target in the hidden cursor condition.

9) The cell count reporting for the visual delay experiments are not transparent. They should be reported in the same way as for visual feedback removal.

We added the requested details on the cell counts for the delay condition to match those reported for the hidden cursor condition. Furthermore, at the request of Reviewer 3, we added a new supplementary figure (Supplementary Figure 1) with additional details on the Purkinje cells recorded and their response characteristics for the parameters evaluated.

Reviewer #3 (Remarks to the Author):

We would also like to thank the Reviewer for their supportive comments stating that “the data and analysis are of high quality and the main results are sufficiently important, to be of wide interest to the community.” We also appreciate the concerns raised that the presentation needs to be substantially revised to make the study more accessible.

Specific points

1) Figure 1 should provide an illustration of the setup and task, together with raw simple spike firing data, its modulation, how it is analyzed in terms of lead and lag, as well as an illustration of the positional error. Panels a and b in Supplementary figure 1 could be used as a basis.

As requested, we added a new Figure 1 that includes all the components suggested.

2) Define all parameters that are used in the text and figures as they arise in the main text. These include Position error XE, YE , $R2, \tau, D\tau$, etc. For several of these the definition in the Methods could be used.

The behavioral parameters are now shown diagrammatically in Figure 1 and defined in the main text in the Results and in the Materials and Methods.

3) Given that the analysis in figure 7 is to control for encoding of kinematics versus prediction error, it seems more logical to refer to the figure early in the text, thereby strengthening the interpretation of the analysis. It could be made a supplementary figure to prevent disruption of the narrative flow.

The Reviewers expressed different views on where to present the analysis of the simple spike modulation with kinematics. The delay and hidden conditions did not affect the simple spike modulation with kinematics. These results demonstrate that the position and velocity modulations are coupled to hand movement and not to the cursor/target. Furthermore, the kinematic results further establish that position error and kinematic signals are independent. We kept the kinematics findings in the main text but near the end of the Results section (Fig. 8).

4) It is not clear why Purkinje cell recordings were made from such a large area (lobules IV-VI). The authors need to be more specific in term recordings location (Zone, zebrin II band) as studies suggest that cerebellar function varies with location. Was there any difference between lobules? Between vermis and paravermis?

Concerning the histology question first, we do not have zebrin II histology nor did we record Purkinje cells in both the vermis and paravermis. In an effort to reduce and refine the use of non-human primates, we target and define our recording locations using a combination of MRI and CT imaging using the software Monkey Cicerone¹³ to register the data followed by anatomical modeling. In both animals, modeling of the recording chamber location and electrode positions show that an electrode in the center of the chamber projects to the intermediate zone close to the primary fissure. We added these details to Materials and Methods.

Second, on the issue of size of the area of the recordings, in the original text we incorrectly reported a larger region than was actually evaluated. In each monkey, the recording grid size was ~ 5 mm x 5 mm in the center of the chamber. Therefore, the recorded area, in the medial-lateral and anterior-posterior dimensions was relatively restricted. Based on the imaging and size of the grid, Purkinje cells were recorded primarily in lobules IV-VI of the intermediate zone. We updated the description of the recording area in the revised manuscript.

5) In relation to point 4, it is clear that the functional properties of Purkinje cells is highly heterogeneous. While this is reported, the authors need to do a better job in describing the criteria they have used to classify responsiveness and it would be instructive to have a visual presentation of the different classes of cells and their frequency. Lastly, they should clearly identify where in the distribution the cells used to illustrate the prediction error and feedback error are found.

Concerning classifying responsiveness, we did not select cells during the recordings. Instead, we report in the manuscript, all Purkinje cell that we correctly identified and successfully recorded through a pair of baseline and feedback manipulations. At the Reviewer's request, a new Supplementary Figure 1 details the number of cells recorded in each pair of conditions and the number with lead and lag modulation with each parameter (position, velocity and position error). As requested by Reviewer 2, we also included more details about the cell counts in the main text.

6) The firing maps are difficult to read, as there are several variables. Is it possible to simplify the presentation so that only the variables of interest are presented? For example, does the direction of the simple spike modulation matter? Could a distance from target center representation be used?

The firing maps are 2-dimensional as each parameter is inherently 2-dimensional. In our opinion, the maps provide the most complete and direct representation of simple spike firing in relation to the behavioral workspaces.

On using a distance measure from the center of the target as the error measure (equivalent to position error magnitude), we had evaluated this measure in a previous paper⁸. Denoted in that study as radial error, the firing of fewer Purkinje cells have a significant fit to radial error than to position error and the strength of the fit to radial error is considerable less than to position error. In addition, by just using distance from the center of the target, the spatial aspect of the simple spike modulation with position error is lost. The example firing plots of simple spike firing clearly show the planar nature of the response with position error (Figs. 3a and 6a). Therefore, we evaluated the simple spike firing with position error as the more appropriate metric.

7) In general, the figures require a clearer explanation for the people not familiar with these types of behavioral experiments.

a. Is the target a big red circle as in figure 1C? If the grey zones in figure S1a and S2a correspond to the moving target, could you make the color match?

We replaced Figure 1C, with a new diagram of the expected results in the hidden condition (Fig. 5). The target path is shown in grey in Figure 1b and c, and the target is a grey circle in Figure 1d and in Figure 5a and c. We used grey as it allows us to show other aspects of the task superimposed on a grey target or path. The actual color of the target in the experiment is red as shown in Figure 1a.

b. Does the black cross in figure 1C correspond to what the monkey sees when he moves the manipulandum?

Yes, the black cross in the new Figure 1d is appropriately scaled to the 3.5 cm target and corresponds to what the animal sees when it moves the cursor.

c. Why do trials have different durations? Are the trials interrupted if the monkey stay out of the area for >700ms?

Each trial consists of the animal tracking one trajectory from a set of 100 trajectories in which the path, speed and duration were varied. Duration was varied to avoid potential confounding factors with temporal expectations due to equal length trials. We clarified these details in the Materials and Methods.

d. Why do cursors have different size? Is it an adjustment specific to the monkey or to the experiment? Is there any protocol for this adjustment?

The differences in the target sizes was an adjustment for the differences in the abilities of the two animals. We clarified the target sizes in the Materials and Methods stating, “because of the differences in each animal’s ability to perform the task with the feedback manipulations, one monkey performed all task conditions with a 2.5 cm target diameter and the other with a 3.5 cm target diameter.”

8) P4 : removing the tracking cursor is described as “reduction of visual feedback” while it is actually more a suppression of all informative visual feedback.

We believe that the reduction in visual feedback is correct, as the cursor is shown when outside the target. Not all informative visual feedback is suppressed in the hidden condition, as the target movement provides some level of visual information (position and speed) to guide the hand movement.

9) P6 and elsewhere there are statements in the text that should have a figure panel citation.

In the revision, we include the appropriate figure panel citations in the text.

10) P6 “the average encoding strength of position error is not affected by delay” seems contradictory with Fig S1c

The Reviewer raises the same concern expressed by Reviewers 1 and 2 that the ANOVA analysis of the probability density plots of position error in the baseline and delay conditions did not reveal any significance difference versus the significant difference obtained with a Student’s t-test performed on position error magnitude. The difference is due to the additional averaging and collapsing the data from the 2 dimensional, 8 x 8 probability density plots to position error magnitude. The effect of these additional steps is to decrease the variability with the position error magnitude being statistically different between the baseline and feedback manipulations.

11) P9 “visual movement of the cursor” is rather vague, and could relate to eye movements or sensory processing of the target. Please be more specific.

We changed this statement to “vision of the cursor movements”.

2) Discussion. The discussion of the number synapses required to generate a simple spike is rather outdated, given that Purkinje cells are intrinsically active even in the absence of inputs in the slice preparation.

We agree with the Reviewer and modified the text to include that the simple spike firing also has an important intrinsic component

13) Discussion p10, In that context, do the authors means "Purkinje cell have the capacity to receive..."

We changed the statement to "Purkinje cells have the bandwidth to carry a large number of independent signals".

14) Discussion. It would be useful to set the current results in the context of recent work in other brain areas showing that expectation mismatch signals are widespread (e.g. Work from Georg Keller in visual cortex). Indeed, could some of the current results, such as the enhanced feedback responses outside the target, be due to the system responding more strongly to unexpected events?

In the Discussion, we added a brief paragraph on the importance of predictions and prediction errors in other regions of the brain.

15) P12. Should "omission of unexpected" read "omission of expected"?

Yes, the statement should read "omission of expected" and has been corrected.

16) Figure 2d,e Green symbols cannot be distinguished from black.

We changed the symbols to better distinguish the black and green symbols. Note that this behavioral data are presented in Supplementary Figure 2.

17) Figure 7. Given basic plots used are similar to those used previously for error, annotation indicating data for velocity are being presented would be useful.

For the simple spike firing plots in revised Figure 8a and f, we added the requested annotation along the y-axes that the results are for velocity.

18) Although it is stated that only lag difference is statistically significant, (Fig6A and 6B & page 8), the figure used for illustrating the lead condition and the much larger error suggest that the lead condition is also affected.

As requested by both Reviewers 1 and 2, we removed these results from the manuscript.

19) Complex spikes are expected to contribute to error encoding. Although they are mentioned in the discussion, they are completely ignored from the paper. It would be relevant to see how they fire compared to simple spike modulation or, at least, what is their contribution is to simple spike firing as they are likely to introduce pauses.

We agree that the role of the complex spikes is important. We are in the process of analyzing the complex spikes. We ask for the reviewer's understanding in not including the complex spike results in this publication, as presenting the complex spike responses and their relation to simple spike modulation will require a stand-alone manuscript. For example, complex spikes do not only encode errors in this task, as detailed in our recent publication of complex spike modulation in baseline pseudo-random tracking¹². In fact, complex spikes modulate more with kinematics than with errors in this task. Also, our recent work shows that complex spikes alter the encoding of information signaled in the simple spikes¹¹.

Concerning the question of the contribution of complex spikes to the simple spike firing, we analyzed the complex spike firing rate in baseline and feedback conditions and the duration of the simple spike pause following the complex spikes for the cells we have evaluated to date. The complex firing rate was stable between baseline and the visual feedback manipulations. The complex spike firing rate was 0.84 ± 0.19 Hz in the baseline vs. 0.78 ± 0.19 Hz in the hidden condition ($t(8)=0.51$, $p = 0.63$) and 0.60 ± 0.29 Hz in baseline vs. 0.51 ± 0.28 Hz in the delay condition ($t(20) = 0.78$, $p = 0.44$). The duration of the simple spike pauses following complex spikes were also stable when comparing baseline to the visual feedback manipulations, Pause duration was 27 ± 35 msec in the baseline vs. 27 ± 30 msec in the hidden condition ($t(6143) = 0.48$, $p = 0.63$) and 37 ± 52 msec in the baseline vs. 37 ± 43 msec in the delay condition ($t(7246) = 0.22$, $p = 0.83$). We conclude that complex spike firing rate was not affected by the visual feedback manipulations nor was the pause in simple spike firing following a complex spike firing.

Reference List

1. Ashe, J. & Georgopoulos, A. P. Movement parameters and neural activity in motor cortex and area 5. **Cereb. Cortex** **4**, 590-600 (1994).
2. Moran, D. W. & Schwartz, A. B. Motor cortical representation of speed and direction during reaching. **J. Neurophysiol.** **82**, 2676-2692 (1999).
3. Paninski, L., Fellows, M. R., Hatsopoulos, N. G., & Donoghue, J. P. Spatiotemporal tuning of motor cortical neurons for hand position and velocity. **J. Neurophysiol.** **91**, 515-532 (2004).
4. Medina, J. F. & Lisberger, S. G. Encoding and decoding of learned smooth pursuit eye movements in the floccular complex of the monkey cerebellum. **J. Neurophysiol.** **102**, 2039-2054 (2009).
5. Shidara, M., Kawano, K., Gomi, H., & Kawato, M. Inverse-dynamics model eye movement control by Purkinje cells in the cerebellum. **Nature** **365**, 50-52 (1993).
6. Gomi, H. *et al.* Temporal firing patterns of Purkinje cells in the cerebellar ventral paraflocculus during ocular following responses in monkeys I. Simple spikes. **J. Neurophysiol.** **80**, 818-831 (1998).

7. Dash, S., Catz, N., Dicke, P. W., & Thier, P. Encoding of smooth-pursuit eye movement initiation by a population of vermal Purkinje cells. **Cereb. Cortex** **22**, 877-891 (2012).
8. Popa, L. S., Hewitt, A. L., & Ebner, T. J. Predictive and feedback performance errors are signaled in the simple spike discharge of individual Purkinje cells. **J. Neurosci.** **32**, 15345-15358 (2012).
9. Hewitt, A., Popa, L. S., Pasalar, S., Hendrix, C. M., & Ebner, T. J. Representation of limb kinematics in Purkinje cell simple spike discharge is conserved across multiple tasks. **J. Neurophysiol.** **106**, 2232-2247 (2011).
10. Popa, L. S., Streng, M. L., & Ebner, T. J. Long-term predictive and feedback encoding of motor signals in the simple spike discharge of Purkinje cells. **eNeuro.** **4**, (2017).
11. Streng, M. L., Popa, L. S., & Ebner, T. J. Climbing fibers control Purkinje cell representations of behavior. **J. Neurosci.** **37**, 1997-2009 (2017).
12. Streng, M. L., Popa, L. S., & Ebner, T. J. Climbing fibers predict movement kinematics and performance errors. **J. Neurophysiol.** **118**, 1888-1902 (2017).
13. Miocinovic, S. *et al.* Stereotactic neurosurgical planning, recording, and visualization for deep brain stimulation in non-human primates. **J. Neurosci. Methods** **162**, 32-41 (2007).

REVIEWERS' COMMENTS:

Reviewer #1 (Remarks to the Author):

The paper is greatly improved. I have no further comments.

Reviewer #2 (Remarks to the Author):

The authors have revised the manuscript so that the motivation and novelty of the study are now clear. Overall, my concerns have been addressed and I have only a few minor presentational suggestions.

Abstract

The motivation of study could be further clarified in the abstract. I recommend inserting a sentence summarizing the groups previous findings in line 3 to state something along the lines, 'Purkinje cell (PCs) simple spiking patterns exhibit both leading and lagging relationships to movement which is consistent with the representation of both motor commands and sensory feedback. This study tested...'

The authors may also want to insert the word 'leading' in line 8: '...demonstrates that the leading simple spike modulation...'

Introduction

The authors should cite Chen et al, 2016 eLife as this paper shows leading and lagging relationships with whisker movements (although not both relationships in single PCs).

It seems noteworthy that both leading and lagging correlations are present in single PCs, and that they therefore can be perturbed independently. Is It worth emphasising this observation at the end of the Introduction? This is already well covered in the Discussion (p.11).

Results

Top of page 5 & Figure 1 – can the authors show example heat plot(s) in this Figure, rather than calling later panels (e.g. Fig 3a)?

Given that similar heat plots are used for position error and kinematics, I recommend adding the label 'Position Error' to panels 3a and 6a analogous to panels 8a,f

I think statistical tests and p values should be stated in main text in addition to legend. Examples include page 6, bottom paragraph and page 8, second paragraph.

Discussion

Page 11: As this appears to be a form of multiplexed coding, the authors should cite Hong et al, 2016 eLife.

Sentence 'Prediction and prediction error processing is considered a unifying principle of brain

function': while I don't necessarily disagree, this statement seems rather overblown as written here.

Reviewer #3 (Remarks to the Author):

The authors have done a good job in making the manuscript simpler, clearer and more accessible. My only remaining comment is that the implications of the heterogeneity in responses observed across cells is not discussed.

Revision of NCOMMS-17-21335A

Response to Reviewers:

We again wish to thank the Reviewers for their helpful comments. We have incorporated every remaining concern and suggestion into the revision. The following are point-by-point responses to the Reviewers' comments.

Reviewer #1 (Remarks to the Author):

The paper is greatly improved. I have no further comments.

We thank Reviewer #1 for their comments. .

Reviewer #2 (Remarks to the Author):

The authors have revised the manuscript so that the motivation and novelty of the study are now clear. Overall, my concerns have been addressed and I have only a few minor presentational suggestions.

We are pleased that the reviewer appreciates the novelty of the study.

Abstract

The motivation of study could be further clarified in the abstract. I recommend inserting a sentence summarizing the groups previous findings in line 3 to state something along the lines, 'Purkinje cell (PCs) simple spiking patterns exhibit both leading and lagging relationships to movement which is consistent with the representation of both motor commands and sensory feedback. This study tested...'

We have modified the Abstract to include a sentence very similar to that suggested by the reviewer. Specifically, "The simple spike firing of cerebellar Purkinje cells both lead and lag movement consistent with representations of motor predictions and sensory feedback." Note that this addition required minor wording changes to the remainder of the Abstract to fit within the 150 word limit.

The authors may also want to insert the word 'leading' in line 8: '....demonstrates that the leading simple spike modulation...'

We have made the requested change to the Abstract, adding "leading".

Introduction

The authors should cite Chen et al, 2016 eLife as this paper shows leading and lagging relationships with whisker movements (although not both relationships in single PCs).

As suggested, the Chen et al, 2016 paper is now cited in the Introduction.

It seems noteworthy that both leading and lagging correlations are present in single PCs, and that they therefore can be perturbed independently. Is It worth emphasizing this observation at the end of the Introduction? This is already well covered in the Discussion (p.11).

We concur with the reviewer and have modified the final statement of the Introduction to emphasize the independence of the predictive and feedback information. Specifically, the last sentence of the Introduction states, "Therefore, the SS activity of individual Purkinje cells independently encodes predictions based on motor commands and sensory feedback about the corresponding behavior."

Results

Top of page 5 & Figure 1 – can the authors show example heat plot(s) in this Figure, rather than calling later panels (e.g. Fig 3a)?

We have included an example heat map of the SS firing in Figure 1f and added the corresponding descriptions in the text and legend.

Given that similar heat plots are used for position error and kinematics, I recommend adding the label 'Position Error' to panels 3a and 6a analogous to panels 8a,f.

We have added "Position Error" to the panels in Figures 3a and 6a.

I think statistical tests and p values should be stated in main text in addition to legend. Examples include page 6, bottom paragraph and page 8, second paragraph.

We have added all appropriate statistical information to the main text as well as retained those statistical details in the legend.

Discussion

Page 11: As this appears to be a form of multiplexed coding, the authors should cite Hong et al, 2016 eLife.

We agree that Purkinje cell simple spike firing multiplexes information and have included the Hong et al, 2016 paper into the statement on Purkinje cell heterogeneity requested by Reviewer #3 (see below).

Sentence 'Prediction and prediction error processing is considered a unifying principle of brain function': while I don't necessarily disagree, this statement seems rather overblown as written here.

We have modified this sentence to be less overstated and now reads, "Therefore, the processing of prediction errors takes place in numerous brain structures and is important for learning and controlling behavior."

Reviewer #3 (Remarks to the Author):

The authors have done a good job in making the manuscript simpler, clearer and more accessible. My only remaining comment is that the implications of the heterogeneity in responses observed across cells is not discussed.

We appreciate that the Reviewer found the revision substantially improved. Concerning the implication of the heterogeneity of the responses across Purkinje cells we have added the following short paragraph to the Discussion:

Similar to our previous studies, individual Purkinje cells encode a combination of position error and kinematics.^{16, 20-22} Further supporting the observation of multiplexing in the cerebellum, SS discharge simultaneously conveys signals on the timing and on-line control of saccadic eye movements.³⁸ Although each Purkinje cell encodes a heterogeneous combination of signals, at the population level there is considerable information about the past and future trajectories of each behavioral variable.^{16, 20} We suggest that this functional heterogeneity provides for a high-fidelity representation of behavior that is flexible enough to compensate for perturbations, as shown in this study by successfully tracking during degraded visual feedback.